# Harnessing Joint Rain-/Detail-aware Representations to Eliminate Intricate Rains

**Wu Ran**[1,2], **Peirong Ma**[1,2], **Zhiquan He**[1,2], **Hao Ren**[1,2], **Hong Lu**[1,2*]
[1]School of Computer Science, Fudan University
[2]Shanghai Key Lab of Intelligent Information Processing
{wran21,zqhe22}@m.fudan.edu.cn,
{prma20,hren17,honglu}@fudan.edu.cn

## ABSTRACT

Recent advances in image deraining have focused on training powerful models on mixed multiple datasets comprising diverse rain types and backgrounds. However, this approach tends to overlook the inherent differences among rainy images, leading to suboptimal results. To overcome this limitation, we focus on addressing various rainy images by delving into meaningful representations that encapsulate both the rain and background components. Leveraging these representations as instructive guidance, we put forth a Context-based Instance-level Modulation (CoI-M) mechanism adept at efficiently modulating CNN- or Transformer-based models. Furthermore, we devise a rain-/detail-aware contrastive learning strategy to help extract joint rain-/detail-aware representations. By integrating CoI-M with the rain-/detail-aware Contrastive learning, we develop CoIC[1], an innovative and potent algorithm tailored for training models on mixed datasets. Moreover, CoIC offers insight into modeling relationships of datasets, quantitatively assessing the impact of rain and details on restoration, and unveiling distinct behaviors of models given diverse inputs. Extensive experiments validate the efficacy of CoIC in boosting the deraining ability of CNN and Transformer models. CoIC also enhances the deraining prowess remarkably when real-world dataset is included.

## 1 INTRODUCTION

Images contaminated with rain can severely impair the performance of outdoor computer vision systems, including self-driving and video surveillance (Wang et al., 2022a). To mitigate the influence of rain, numerous image deraining methods have emerged over the past decades with the objective of restoring pristine backgrounds from their rain-corrupted counterparts. Recent years have witnessed the notable success of the learning-based methods (Fu et al., 2017; Li et al., 2018; Yang & Lu, 2019; Zamir et al., 2021; Mou et al., 2022; Zamir et al., 2022; Xiao et al., 2022; Özdenizci & Legenstein, 2023), which leverage large labeled datasets to train sophisticated image deraining models.

A number of contemporary learning-based methods (Yang et al., 2017; Li et al., 2018; Yang et al., 2019; Xiao et al., 2022) exclusively train and validate models on a single dataset. However, such a strategy is infeasible for practical applications, as the requisite training time and physical storage scale linearly with the number of distinct datasets. Furthermore, a synthetic dataset tend to exhibit restricted diversity in rain characteristics like orientation, thickness, and density, as it is generated utilizing a unitary simulation technique, *e.g.*, photorealistic rendering (Garg & Nayar, 2006), physical modeling (Li et al., 2019), and Photoshop simulation[2]. As a consequence, models trained on a single dataset frequently generalize poor to others. Recent work (Zamir et al., 2021; 2022; Mou et al., 2022; Wang et al., 2023) has explored training models on amalgamated datasets drawn from multiple sources, yielding enhanced performance under adverse rainy conditions while avoiding overfitting to specific datasets. Nevertheless, these methods directly mix all datasets, which risks

---

*Corresponding author
[1]Code is available at: https://github.com/Schizophreni/CoIC
[2]Rain rendering: https://www.photoshopessentials.com/photo-effects/rain/

neglecting the discrepancies among datasets and resulting in suboptimal optimization. As illustrated in Figure 1 (a), rain density across mixed datasets exhibits a long-tail distribution spanning a wide range. Consequently, models directly trained on mixed datasets with suboptimal optimization may exhibit poor real-world deraining ability, as illustrated in Figure 1 (c).

To address the aforementioned limitations, we propose to learn adaptive image deraining through training on mixed datasets. The goal is to exploit the *commonalities* and *discrepancies* among datasets for training. Specifically, the model architecture and base parameters are shared (commonalities), while image representations are extracted to modulate the model's inference process (discrepancies). These representations provide instructive guidance for a novel Context-based Instance-level Modulation (CoI-M) mechanism, which can efficiently modulates both CNN and Transformer architectures. CoI-M is also verified to improve the performances of existing models trained on mixed datasets.

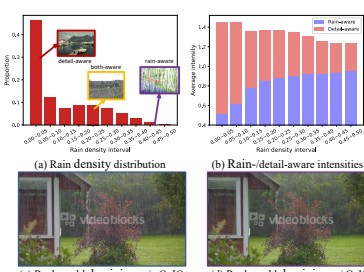

(a) Rain density distribution    (b) Rain-/detail-aware intensities

(c) Real-world deraining w/o CoIC    (d) Real-world deraining w/ CoIC

Further analysis reveals that all these rain-/detail-aware representations form a unified and meaningful embedding space, where images with light rain are primarily characterized by background detail, while heavy rainy images are distinguished more by the rain itself. A statistical analysis in Figure 1 (b) indicates that the embedding becomes increasingly rain-aware and less detail-aware as rain density increases. This suggests that gathering both background and rain factors becomes crucial when rain density spans a wide range,

Figure 1: (a) rain density distribution. (b) rain-/detail-awareness intensities with respect to rain density. (c) & (d) real-world deraining results of DGUNet (Mou et al., 2022) trained on three mixed datasets without and with the proposed CoIC, respectively.

which is neglected in previous works (Li et al., 2022a; Ye et al., 2022). These observations motivate learning a meaningful joint embedding space that perceives various rains and background details.

Contrastive learning has been widely adopted to learn image representations in an unsupervised manner. He et al. (2020) propose a content-related instance-discriminative contrastive learning algorithm, where the degradation factor is overlooked. More recently, contrastive learning-based image restoration approaches have sprung up. Wang et al. (2021) assume that degradations of two images are different. However, such an assumption may be infeasible for rainy images when multiple datasets are mixed. Li et al. (2022a) focus on learning discriminative rain, fog, and snow representations. Ye et al. (2022) attempt to separate rain layer from background. Chen et al. (2022) propose a dual contrastive learning approach to push rainy and clean images apart. Nevertheless, (Li et al., 2022a; Ye et al., 2022; Chen et al., 2022) all fail to perceive coupled intricate rains and background details. Wu et al. (2023) introduce contrastive learning as a perceptual constraint for image super-resolution, but it can only distinguish between degraded and high-quality images. Therefore, learning joint rain-/detail-aware representations serves as a non-trivial problem, considering diverse coupling modes of rain and background among datasets. To this end, we propose a novel joint rain-/detail-aware contrastive learning approach. Further, by integrating it with CoI-M, we develop CoIC, a strategy to train high-performance, generalizable CNN or Transformer models using mixed datasets. As illustrated in Figure 1 (c) & (d), models trained under the CoIC strategy demonstrate improved deraining ability to real-world rainy images.

**Contributions**. In this work, we propose to learn adaptive deraining models on mixed datasets through exploring joint rain-/detail-aware representations. The key contributions are as follows: (1). We introduce a novel context-based instance-level modulation mechanism for learning adaptive image deraining models on mixed datasets. Rain-/detail-aware representations provide instructive guidance for modulation procedure. (2). To extract rain-/detail-aware representations effectively, we develop a joint rain-/detail-aware contrastive learning strategy. This strategy facilitates the learning of high-quality representations for CoI-M. (3). By integrating CoI-M and the proposed contrastive learning strategy, we introduce the CoIC to enhance deraining performance for models trained on mixed datasets. CoIC provides insight into exploring the relationships between datasets and enables quantitative assessment of the impact of rain and image details. Experimental results demonstrate that CoIC can significantly improve the performance of CNN and Transformer models, as well as enhance their deraining ability on real rainy images by including real-world dataset for training.

## 2 RELATED WORK

**Image Deraining**. Traditional image deraining methods focus on separating rain components by utilizing carefully designed priors, such as Gaussian mixture model (Li et al., 2016), sparse representation learning (Gu et al., 2017; Fu et al., 2011), and directional gradient prior (Ran et al., 2020). Generally, these methods based on hand-crafted priors tend to lack generalization ability and impose high computation burdens. With the rapid development of deep learning, various deep neural networks have been explored for image deraining. Fu et al. (2017) propose the pioneering deep residual network for image deraining. Subsequent methods by Li et al. (2018); Yang & Lu (2019); Ren et al. (2019) incorporate recurrent units to model accumulated rain. Li et al. (2019) introduce depth information to remove heavy rain effects. Considering the cooperation between synthetic and real-world data, Yasarla et al. (2020) propose a semi-supervised approach. More recently, Ye et al. (2022); Chen et al. (2022) propose contrastive learning-based unsupervised approaches. Additionally, Xiao et al. (2022); Chen et al. (2023) have designed effective transformer models. Note that a majority of deraining models are trained on individual datasets, restricting them to adverse rain types and background scenes. Thus, some recent work (Zamir et al., 2021; Mou et al., 2022; Zamir et al., 2022; Wang et al., 2023) leverage multiple datasets to train more robust models. However, simply mixing and training on amalgamated datasets overlooks potential discrepancies among them, consequently resulting in suboptimal results. To address this issue, we propose to learn adaptive image deraining models by exploring meaningful rain-/detail-aware representations. The proposed approach helps models efficiently capture both commonalities and discrepancies across multiple datasets.

**Image Restoration with Contrastive Learning**. Contrastive learning has emerged as an efficient approach for unsupervised representation learning (Chen et al., 2020a;b; He et al., 2020). The underlying philosophy is to pull similar examples (positives) together while pushing dissimilar examples (negatives) apart in the latent feature space. Some recent work (Wu et al., 2021; Li et al., 2022b; Fu et al., 2023) has employed contrastive learning as auxiliary constraints for model training. Ye et al. (2022); Chen et al. (2022) propose unsupervised deraining methods by pushing rain and clean components apart. Wang et al. (2021) and Li et al. (2022a) focus on extracting degradation representations for image super-resolution and all-in-one restoration, respectively. Most recently, Zheng et al. (2024) employ contrastive learning to generate realistic rainy images by controlling the amount of rain. These *off-the-shelf* approaches either learn only degradation representations or only discriminate between rain and clean background, which cannot perceive coupled rain and backgrounds.

**Image Restoration with Model Modulation**. He et al. (2019) inserts adaptive modulation modules to control image restoration on continuous levels. Fan et al. (2019) proposes to generate image operator parameters with hand-crafted guidances. Recently, Li et al. (2022a) introduce deformable convolution into image restoration, where its offset and scales are generated with deep feature guidance. Inspired by (Hu et al., 2021), we propose a computation-friendly context-based modulation mechanism using vector-level guidance.

## 3 METHODOLOGY

### 3.1 PROBLEM DEFINITION

Given mixed multiple rainy datasets $\mathcal{D} = \cup_{i=1}^{M} D_i$, our objective is to learn the optimal parameters $\theta^*$ for a model that minimizes the overall empirical cost across datasets, *i.e.*, $\arg\min_\theta \ell(\theta; \mathcal{D})$. Notably, $M$ is the number of datasets and $\ell$ denotes any arbitrary loss function. A straightforward approach, as proposed in recent work (Jiang et al., 2020; Zamir et al., 2021; 2022; Wang et al., 2023), is to directly train the model on mixed datasets. We argue that such an approach risks neglecting the discrepancies among datasets, *e.g.*, the rain types, the background scenes, and the rain simulation techniques, resulting in suboptimal performance. Moreover, directly training on mixed datasets impedes the model's ability to assimilate knowledge across datasets. Therefore we propose an adaptive image deraining approach, termed CoIC, to facilitate training models on mixed datasets. An overview of the proposed CoIC is presented in Figure 2 (a). In particular, a unified and structural latent embedding space $\mathcal{E}$ for $\mathcal{D}$ is constructed to capture rain-/detail-aware properties for all images. Formally, we aim to optimize $\ell(\theta, \mathcal{E}; \mathcal{D})$. For simplicity, we decompose it by:

$$\ell(\theta, \mathcal{E}; \mathcal{D}) = \ell\left(\theta(\mathcal{E}); \mathcal{D}\right) + \lambda \mathcal{P}(\mathcal{E}; \mathcal{D}), \tag{1}$$

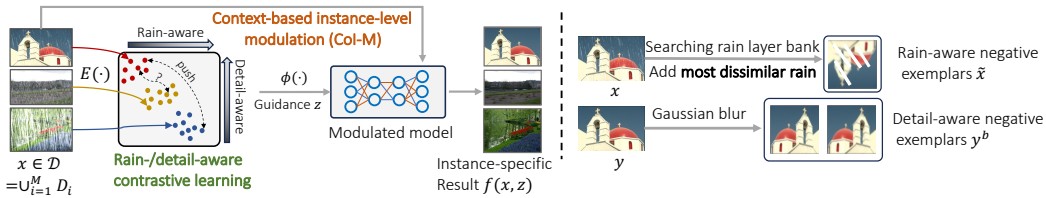

(a) An overview of the proposed CoIC  (b) Rain-/detail-aware negative exemplars generation

Figure 2: (a) The framework of the proposed CoIC. We extract instance-level representations with the help of rain-/detail-aware contrastive learning strategy. Leveraging these representations as instructive guidance, we then utilize CoI-M to modulate the model's parameters, yielding adaptive deraining results. (b) Generation of rain-/detail-aware negative exemplars.

where the first term is a data-fidelity term, and the second term indicates an embedding space constraint. $\lambda$ is a hyper-parameter which is heuristically tuned in Appendix A.4.

## 3.2 INSTANCE-LEVEL REPRESENTATION EXTRACTION

Let $\mathcal{D} = \{(x_j, y_j)\}_{j=1}^{N}$ denote the mixed multiple rainy dataset, where $x_j$ and $y_j$ are the paired rainy input and ground truth, respectively, and $N$ is the total number of pairs. To obtain the rain-/detail-aware representation, we employ an encoder comprising a feature extractor $E$, a Global Average Pooling (GAP) operation, and a subspace projector $\phi$ to embed the rainy image $x \in \mathbb{R}^{H \times W \times 3}$ into a $d$-dimensional spherical vector $z \in \mathbb{R}^d$ (see Figure 3 for the details). Applying $E$ to $x$ produces a feature map $F = E(x)$ which captures rich spatial and channel information related to rain and image details. *Considering the property in F, we introduce a contrastive learning strategy, detailed in Section 3.3, to learn a rain-/detail-aware embedding space.* We then project the feature map $F$ into a $d$-dimensional vector as shown in Figure 3 by

$$z = \frac{\phi\left(\text{GAP}(F)\right)}{||\phi\left(\text{GAP}(F)\right)||_2}, \tag{2}$$

where $\phi$ indicates the subspace projector. As a result, the latent vector $z$ encapsulates rain-/detail-aware information, which can be leveraged to guide the adaptive image deraining.

## 3.3 JOINT RAIN-/DETAIL-AWARE CONTRASTIVE LEARNING

As previously discussed, the embedding space $\mathcal{E}$ (see in Figure 2 (a)) characterizes both rain and image details. Concretely, the encoder is required to focus more on image details for light rain images while concentrating more on rain for heavy rain images (see in Figure 1 (b)).

To achieve this, we develop a rain-/detail-aware contrastive learning approach by carefully designing negative exemplars (see Figure 2 (b)).

**Negative Exemplars in Rain-aware Case**. Given a rainy-clean image pair $(x, y)$, the encoder should discriminate the rain in $x$. To this end, we leverage a *rain layer bank* noted as $D_R = \{r_1, r_2, \cdots r_u\}$ to generate negative exemplar as follows. we first retrieve $D_R$ to obtain a rain layer that is *most dissimilar* to the rain in $x$, which is determined by

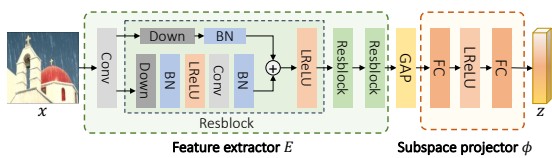

Figure 3: The encoder for extracting instance-level representations comprises three components: a feature extractor, GAP layer, and subspace projector. LReLU indicates the LeakyReLU activation.

$$\tilde{r} = \arg \max_{r \in D_R} ||(x - y) - r||_1. \tag{3}$$

Utilizing $\tilde{r}$, we can generate a negative exemplar $\tilde{x} = y + \tilde{r}$, which contains the most dissimilar rain to $x$ but preserving the same background. In practice, the rain layer bank is constructed from the data batch, where *cross-dataset simulation* is facilitated.

**Negative Exemplars in Detail-aware Case**. Recently, Wu et al. (2023) has developed an efficient contrastive learning-based perceptual loss for image super-resolution, where the blurred editions of input image are considered as negative exemplars. Motivated by this, the detail-aware information can be exploited through distinguishing between the details in rainy image from the blurred clean image. Specifically, given a rainy-clean pair $(x, y)$, we generate $N_b$ negative exemplars by blurring the clean image $y$ to obtain $\{y_j^b\}_{j=1}^{N_b}$.

With the negative exemplars $\tilde{x}$ and $\{y_j^b\}$, we formulate the proposed contrastive learning loss as

$$\ell_{\text{contra}} = -\log \frac{e^{\cos(F, F_k)}}{e^{\cos(F, F_k)} + e^{\cos(F, \tilde{F})} + \sum_{j=1}^{N_b} e^{\cos(F, F_j^b)}}, \tag{4}$$

where $F = E(x)$, $F_k = \tilde{E}(x^k)$, $\tilde{F} = \tilde{E}(\tilde{x})$, and $F_j^b = \tilde{E}(y_j^b)$ are spatial features extracted by feature extractor $E$ and its momentum-updated version $\tilde{E}$ (following MoCo (He et al., 2020)), containing rich rain-/detail-aware information. $\cos(\cdot, \cdot)$ denotes channel-wise cosine similarity. Here, $x^k$ denotes an augmentation of $x$. As a result, $\ell_{\text{contra}}$ forms the embedding space constraint term in equation 1, *i.e.*, the $\mathcal{P}(\mathcal{E}; \mathcal{D})$. The rain-/detail-aware information in $F$ will propagate to the vector $z$ via equation 2, which can guide the modulation for both the CNN and Transformer models.

**How to Assess the Impact of Rain and Details**? As stated previously, the feature extractor $E$ effectively characterizes both rain- and detail-related properties, enabling an assessment of the model's reliance on rain cues versus background details. Specifically, given a rainy-clean pair $(x, y)$, the dependance on background details can be qualified as $\zeta_B = \cos(\text{GAP}(E(x)), \text{GAP}(E(y)))$, where a higher $\zeta_B$ indicates greater dependence on details. To compute $\zeta_R$ for rain effect, we build a *clean image bank* denoted as $D_B$ and use it to retrieve the *most dissimilar* background $\tilde{b}$ by

$$\tilde{b} = \arg \max_{b \in D_B} ||y - b||_1. \tag{5}$$

Subsequently, we construct the image $x' = \tilde{b} + (x - y)$ which retains the rain in $x$ while incorporating maximally dissimilar background. Formally, $\zeta_R$ is defined as $\cos(\text{GAP}(E(x)), \text{GAP}(E(x')))$. A higher $\zeta_R$ indicates greater reliance on rain. Figure 1 (b) provides a rain-/detail-awareness analysis utilizing $\zeta_B$ and $\zeta_R$.

**Discussion**. The Gaussian blur may not effectively model the degraded background in excessively rainy images where occlusions dominate. However, in such case, the rain-aware information typically dominates, thereby mitigating the limitations of Gaussian blur.

### 3.4 CONTEXT-BASED INSTANCE-LEVEL MODULATION

Leveraging the rain/detail-related information encapsulated in $z$ from equation 2, we propose a Context-based Instance-level Modulation (CoI-M) mechanism to modulate each convolutional layer in a CNN model and self-attention layer in a Transformer model under the guidance of $z$.

**Modulate CNN-based Model**. The convolutional layer is the fundamental building block of CNN models, comprised of $C$ filters each with dimensions $C' \times k \times k$. Notably, $C'$ and $C$ denote the number of input and output channels, while $k$ refers to the spatial size of the kernel, respectively. Let the filters in the $l$-th convolutional layer be represented by $\mathbf{W}^l \in \mathbb{R}^{C_l \times C_l' \times k \times k}$. Given an input feature $F^l$, the standard convolution operation yields

$$F^{l+1} = \mathbf{W}^l \star F^l + b^l, \tag{6}$$

where $\star$ denotes the convolutional operation and $b^l$ is a bias. We aim to modulate the output $F^{l+1}$ by modulating $\mathbf{W}^l$ under the guidance of instance-level embedding $z$ and the context in $F^l$. Inspired by the low-rank adaptation in large language models (Hu et al., 2021), we generate an adaptive weight $\mathbf{A}^l \in \mathbb{R}^{C_l \times C_l' \times k \times k}$ to modulate $\mathbf{W}^l$. Spefically, we first generate three vectors utilizing $z$ and the context in $F^l$ following

$$[Q^l, R^l, Z^l] = \text{MLP}([z, \text{GAP}(F^l)]), \tag{7}$$

where $Q^l \in \mathbb{R}^{C_l}$, $R^l \in \mathbb{R}^{C_l'}$, and $Z^l \in \mathbb{R}^{k \times k}$ (by reshaping) are cropped from the output of MLP, and MLP is a two-layer multi-layer perceptron. Then we generate $\mathbf{A}^l$ by

$$\mathbf{A}_{c,c',\alpha,\beta}^l = \frac{k^2 e^{Z_{\alpha,\beta}^l / \tau_{cc'}^l}}{\sum_{\alpha',\beta'} e^{Z_{\alpha',\beta'}^l / \tau_{cc'}^l}}, \tag{8}$$

where $\tau_{cc'}^l = 1/\text{sigmoid}(Q_c^l + R_{c'}^l)$. Note that $\tau_{cc'}$ can induce a channel-wise temperature $T_{cc'}^l = k^2\tau_{cc'}/\sum_{\alpha,\beta}(Z_{\alpha,\beta}^l - \min Z_{\alpha,\beta}^l)$ of a standard Softmax operation in equation 8 (Proof and an in-depth analysis can be found in Appendix A.1), which controls the receptive filed of a $k \times k$ convolutional kernel. Moreover, the nonlinearity in equation 8 can efficiently increase the rank of $\mathbf{A}^l$. Utilizing the adaptive weight $\mathbf{A}^l$, we can modulate the convolution operation in equation 6 via

$$\tilde{F}^{l+1} = (\mathbf{A}^l\mathbf{W}^l) \star F^l + b^l = F^{l+1} + \Delta\mathbf{W}^l \star F^l, \tag{9}$$

where the last term indicates an adaptive alternation to $F^{l+1}$ in equation 6, with $\Delta\mathbf{W}^l = (\mathbf{A}^l - 1)\mathbf{W}^l$. Furthermore, if all elements in $Z^l$ are equal, we can derive from equation 8 and 9 that $\Delta\mathbf{W}^l = 0$ ($T_{cc'}^l \to \infty$), implying no modulation under this condition. This modulation approach thereby enables efficient modulation of the CNN models under the guidance of instance-level representation $z$ and the feature context. In practice, *equation 9 can be efficiently computed in parallel utilizing group convolution.* A PyTorch-style implementation is provided in Appendix A.2.

**Modulate Transformer-based Architecture**. In Transformer-based models, where the self-attention layer is a core building block, we propose incorporating the instance-level representation $z$ to develop a cross-attention mechanism as follows:

$$\mathbf{Q}, c = X\mathbf{W}^Q, \text{MLP}(z),$$
$$\mathbf{K}, \mathbf{V} = (X + c)\mathbf{W}^K, (X + c)\mathbf{W}^V,$$
$$Y = \text{Softmax}\left(\frac{\mathbf{Q}\mathbf{K}^T}{\sqrt{d_k}}\right)\mathbf{V}, \tag{10}$$

where $X$ and $Y$ denote the input feature and output cross-attention result, respectively. In particular, $c$ represents the context generated from representation $z$ in equation 2. MLP refers to a SiLU activation followed by a fully-connected layer. Cross-attention mechanism has been efficiently adopted to image translation tasks (Rombach et al., 2022). Owing to the cross-attention formulation in equation 10, the model can efficiently generate adaptive deraining results guided by the instance-level representation $z$. Similar to equation 9, the modulation of $\mathbf{K}$ (or $\mathbf{V}$) in equation 10 can be formulated as $\mathbf{K} = X\mathbf{W}^K + X\Delta\mathbf{W}^K$, where $X\Delta\mathbf{W}^K$ can be equivalently transferred into $\Delta X\mathbf{W}^K$. The alternation $\Delta X = \text{MLP}(z)$ thus is equivalent to equation 10.

The proposed CoI-M facilitates adaptive deraining in a layer-wise modulated manner. Let $f(x, z)$ denote the restored image under the guidance of $z$. The data-fidelity loss term in equation 1 can be derived below:

$$\ell(\theta(\mathcal{E}); \mathcal{D}) = \mathbb{E}_{(x,y)\sim\mathcal{D}}\ell\left(f(x, z), y\right), \ \mathcal{E} = \text{span}(\{z\}), \tag{11}$$

where $\ell(\cdot, \cdot)$ represents arbitrary data-fidelity loss function or a combination of multiple loss functions, *e.g.*, MSE or $L_1$ loss, and $\mathcal{E}$ denotes the joint rain-/detail-aware embedding space spanned by all instance-level representations. The in-depth analyses of the embedding space can be found in Section 4.2 and Appendix A.6.

# 4 EXPERIMENTAL RESULTS

## 4.1 SETTINGS

**Synthetic and Real-world Datasets**. We conduct extensive experiments utilizing five commonly adopted synthetic datasets: Rain200L & Rain200H (Yang et al., 2017), Rain800 (Zhang et al., 2019), DID-Data (Zhang & Patel, 2018), and DDN-Data (Fu et al., 2017). Rain200L and Rain200H contain light and heavy rain respectively, each with 1800 image pairs for training and 200 for evaluation. Rain800 is generated using Photoshop[3] with diverse and accumulated rain types varying in orientation, intensity, and density. It has 700 pairs for training and 100 for testing. DID-Data generated utilizing Photoshop comprises three rain density levels, each with 4000/400 pairs for training/testing. DDN-Data consists of 12,600 training and 1400 testing pairs with 14 rain augmentations. In total, we amalgamate all 5 training sets comprising 28,900 pairs as the mixed training set (much larger than current mixed dataset Rain13K (Jiang et al., 2020)). To evaluate the real-world deraining ability, we use the real-world dataset from (Wang et al., 2019) comprising 146 challenging rainy images, which we denote as RealInt.

---

[3]http://www.photoshopessentials.com/photo-effects/rain/

Table 1: Quantitative comparison of five representative models trained on mixed multiple datasets. The last column demonstrates the real-world deraining quality on RealInt.

| Methods | Rain200L | | Rain200H | | Rain800 | | DID-Data | | DDN-Data | | RealInt |
|---|---|---|---|---|---|---|---|---|---|---|---|
| | PSNR ↑ | SSIM ↑ | PSNR ↑ | SSIM ↑ | PSNR ↑ | SSIM ↑ | PSNR ↑ | SSIM ↑ | PSNR ↑ | SSIM ↑ | NIQE ↓ |
| Syn2Real (Yasarla et al., 2020) | 30.83 | 0.9386 | 17.21 | 0.5554 | 24.85 | 0.8478 | 26.71 | 0.8759 | 29.15 | 0.9033 | 4.9052 |
| DCD-GAN (Chen et al., 2022) | 21.64 | 0.7734 | 16.04 | 0.4782 | 19.52 | 0.7717 | 21.28 | 0.8059 | 21.60 | 0.8020 | 4.7640 |
| BRN (Ren et al., 2020) | 35.81 | 0.9734 | 27.83 | 0.8819 | 24.15 | 0.8632 | 33.52 | 0.9515 | 32.40 | 0.9441 | 4.7008 |
| BRN + CoIC (**Ours**) | $37.81_{\uparrow 2.00}$ | $0.9816_{\uparrow 0.0082}$ | $28.43_{\uparrow 0.60}$ | $0.8903_{\uparrow 0.0084}$ | $26.13_{\uparrow 1.98}$ | $0.8839_{\uparrow 0.0207}$ | $34.01_{\uparrow 0.49}$ | $0.9539_{\uparrow 0.0024}$ | $32.92_{\uparrow 0.52}$ | $0.9476_{\uparrow 0.0035}$ | $4.5963_{\downarrow 0.1045}$ |
| RCDNet (Wang et al., 2020) | 36.73 | 0.9737 | 28.11 | 0.8747 | 25.29 | 0.8626 | 33.65 | 0.9516 | 32.88 | 0.9451 | 4.8781 |
| RCDNet + CoIC (**Ours**) | $37.63_{\uparrow 0.90}$ | $0.9779_{\uparrow 0.0042}$ | $29.13_{\uparrow 1.02}$ | $0.8858_{\uparrow 0.0111}$ | $26.44_{\uparrow 1.15}$ | $0.8847_{\uparrow 0.0221}$ | $33.98_{\uparrow 0.33}$ | $0.9525_{\uparrow 0.0009}$ | $33.05_{\uparrow 0.17}$ | $0.9462_{\uparrow 0.0011}$ | $4.8168_{\downarrow 0.0613}$ |
| DGUNet (Mou et al., 2022) | 39.80 | 0.9866 | 31.00 | 0.9146 | 30.43 | 0.9088 | 35.10 | 0.9624 | 33.99 | 0.9555 | 4.7040 |
| DGUNet + CoIC (**Ours**) | $39.88_{\uparrow 0.08}$ | $0.9868_{\uparrow 0.0002}$ | $31.07_{\uparrow 0.07}$ | $0.9152_{\uparrow 0.0006}$ | $30.75_{\uparrow 0.32}$ | $0.9183_{\uparrow 0.0095}$ | $35.11_{\uparrow 0.01}$ | $0.9627_{\uparrow 0.0003}$ | 33.99 | $0.9556_{\uparrow 0.0001}$ | $4.6008_{\downarrow 0.1032}$ |
| IDT (Xiao et al., 2022) | 39.37 | 0.9857 | 30.04 | 0.9115 | 28.71 | 0.9093 | 34.62 | 0.9609 | 33.56 | 0.9532 | 4.6308 |
| IDT + CoIC (**Ours**) | $39.76_{\uparrow 0.39}$ | $0.9865_{\uparrow 0.0008}$ | $30.58_{\uparrow 0.54}$ | $0.9173_{\uparrow 0.0058}$ | $29.20_{\uparrow 0.49}$ | $0.9111_{\uparrow 0.0018}$ | $34.91_{\uparrow 0.29}$ | $0.9626_{\uparrow 0.0017}$ | $33.77_{\uparrow 0.21}$ | $0.9548_{\uparrow 0.0016}$ | $4.6080_{\downarrow 0.0228}$ |
| DRSformer (Chen et al., 2023) | 39.74 | 0.9858 | 30.42 | 0.9057 | 29.86 | 0.9114 | 34.96 | 0.9607 | 33.92 | 0.9541 | 4.7562 |
| DRSformer + CoIC (**Ours**) | $39.81_{\uparrow 0.07}$ | $0.9862_{\uparrow 0.0004}$ | $30.50_{\uparrow 0.08}$ | $0.9076_{\uparrow 0.0019}$ | $29.92_{\uparrow 0.06}$ | $0.9115_{\uparrow 0.0001}$ | $35.01_{\uparrow 0.05}$ | $0.9614_{\uparrow 0.0007}$ | $33.94_{\uparrow 0.02}$ | $0.9545_{\uparrow 0.0004}$ | $4.6593_{\downarrow 0.0969}$ |

Table 2: Quantitative results of further trained DRSformer with SPAData.

| Methods | Rain200L | | Rain200H | | Rain800 | | DID-Data | | DDN-Data | | SPAData | |
|---|---|---|---|---|---|---|---|---|---|---|---|---|
| | PSNR ↑ | SSIM ↑ | PSNR ↑ | SSIM ↑ | PSNR ↑ | SSIM ↑ | PSNR ↑ | SSIM ↑ | PSNR ↑ | SSIM ↑ | PSNR ↑ | SSIM ↑ |
| DRSformer | 39.32 | 0.9850 | 29.27 | 0.9000 | 28.85 | 0.9070 | 34.91 | 0.9607 | 33.71 | 0.9540 | 45.46 | 0.9898 |
| DRSformer + CoIC (**Ours**) | $39.70_{\uparrow 0.38}$ | $0.9860_{\uparrow 0.0010}$ | $30.31_{\uparrow 1.04}$ | $0.9058_{\uparrow 0.0058}$ | $29.73_{\uparrow 0.88}$ | $0.9143_{\uparrow 0.0073}$ | $35.02_{\uparrow 0.11}$ | $0.9618_{\uparrow 0.0011}$ | $33.94_{\uparrow 0.23}$ | $0.9556_{\uparrow 0.0016}$ | $46.03_{\uparrow 0.57}$ | $0.9903_{\uparrow 0.0005}$ |

**Evaluation Metrics**. Following previous work (Zamir et al., 2021; 2022), we adopt two commonly used quantitative metrics for evaluations: Peak Signal-to-Noise Ratio (PSNR) (Huynh-Thu & Ghanbari, 2008) and Structural Similarity Index (SSIM) (Wang et al., 2004). For real-world images, we utilize the Natural Image Quality Evaluator (NIQE) metric (Mittal et al., 2012).

**Training Settings**. The base channel number in the feature extractor is set to 32. After each downsampling operation, the channel number is doubled. All LeakyReLU layers in the feature extractor have a negative slope of $0.1$. The output dimension of the subspace projector is 128, corresponding to the dimension of $z$ in equation 2, following (He et al., 2020). For rain-/detail-aware contrastive learning, the number of detail-aware negative exemplars is set to $N_b = 4$ as suggested in (Wu et al., 2023). The blurred negative exemplars are generated using Gaussian blur with sigma uniformly sampled from interval $[0.3, 1.5]$. The hyper-parameter $\lambda$ balancing the contribution of the contrastive loss in equation 1 is empirically set to $0.2$. Experiments are implemented in PyTorch (Paszke et al., 2019) on NVIDIA GeForce GTX 3090 GPUs.

## 4.2 MAIN RESULTS

**Comparison on Benchmark Datasets**. We first verify the effectiveness of CoIC through training models on a mixture of five synthetic datasets. Specifically, three recent CNN models are selected, including BRN (Ren et al., 2020), RCDNet (Wang et al., 2020), and DGUNet (Mou et al., 2022), along with two Transformer models, IDT (Xiao et al., 2022) and DRSformer (Chen et al., 2023). A model complexity analysis, along with training details and training histories are provided in Appendix A.3 & A.5, respectively. Additionally, two representative semi-supervised method Syn2Real (Yasarla et al., 2020) and contrastive learning-based unsupervised method DCD-GAN (Chen et al., 2022) are included for comparsion. The evaluation results are tabulated in Table 1. Due to the complexity of rain and backgrounds, both Syn2Real and DCD-GAN fail with unsatisfactory performances. It can be seen that CoIC has brought significant performance improvements across all synthetic datasets, while also exhibiting superior real-world deraining capabilities. This provides evidence that the proposed CoIC approach can substantially enhance deraining performance when training models on mixed datasets. A visual comparison is included in Appendix A.7. Note that Mixture of Experts (MoE) modules in DRSformer help it tolerate discrepancies across synthetic datasets, resulting in marginal improvement. However, DRSformer faces significant challenges when incorporating a real-world dataset for further training (please refer to Table 2).

To further investigate the efficacy of the proposed CoIC when training models on individual datasets, experiments are conducted with 10 representative methods: DDN (Fu et al., 2017), DID-MDN (Zhang & Patel, 2018), JORDER-E (Yang et al., 2019), PReNet (Ren et al., 2019), RCD-Net (Wang et al., 2020), DGCN (Fu et al., 2021), SPDNet (Yi et al., 2021), Restormer (Zamir et al., 2022), Uformer (Wang et al., 2022b), and DRSformer (Chen et al., 2023). The heavy rain dataset, Rain200H is chosen for benchmarking. Quantitative results are presented in Table 3, where we find that DRSformer with CoIC has achieved the best PSNR metric compared to other methods, offer-

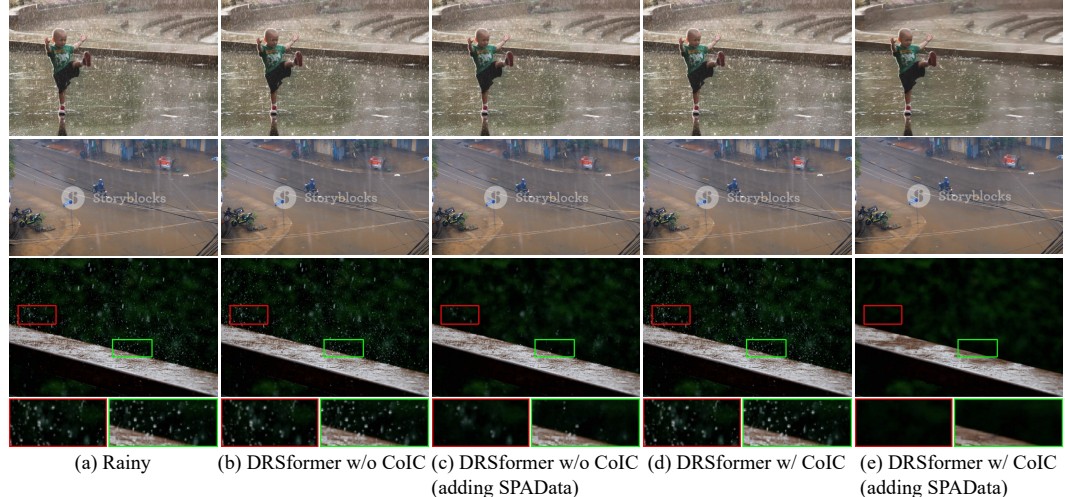

| (a) Rainy | (b) DRSformer w/o CoIC | (c) DRSformer w/o CoIC (adding SPAData) | (d) DRSformer w/ CoIC | (e) DRSformer w/ CoIC (adding SPAData) |

Figure 4: Real-world image deraining comparsion on challenging rainy images from RealInt.

ing 0.09dB PSNR gain over DRSformer. This indicates that the proposed CoIC can also improve deraining performance on individual datasets.

Table 3: Quantitative comparison of image deraining methods on Rain200H dataset.

| Methods | Rain200H PSNR↑ | Rain200H SSIM↑ |
|---|---|---|
| DDN (Fu et al., 2017) | 26.05 | 0.8056 |
| DID-MDN (Zhang & Patel, 2018) | 26.61 | 0.8242 |
| JORDER-E (Yang et al., 2019) | 29.35 | 0.8905 |
| PReNet (Ren et al., 2019) | 29.04 | 0.8991 |
| RCDNet (Wang et al., 2020) | 30.24 | 0.9048 |
| DGCN (Fu et al., 2021) | 31.15 | 0.9125 |
| SPDNet (Yi et al., 2021) | 31.28 | 0.9207 |
| Restormer (Zamir et al., 2022) | 32.00 | 0.9329 |
| Uformer (Wang et al., 2022b) | 30.80 | 0.9105 |
| DRSformer (Chen et al., 2023) | 32.17 | 0.9326 |
| DRSformer + CoIC (**Ours**) | **32.26** | **0.9327** |

Table 4: Comparison on the RealInt dataset. Restormer[†] denotes the official pre-trained model. Values marked with ↓ specify the NIQE gain.

| Model | Mix | NIQE↓ |
|---|---|---|
| Restormer[†] (Zamir et al., 2022) | ✓ | 4.8498 |
| DGUNet (Mou et al., 2022) | ✗ | 4.7373 |
| DGUNet (Mou et al., 2022) | ✓ | 4.7040 |
| DGUNet + CoIC (**Ours**) | ✓ | **4.6008**$_{\downarrow 0.1032}$ |
| IDT (Xiao et al., 2022) | ✗ | 5.5352 |
| IDT (Xiao et al., 2022) | ✓ | 4.6308 |
| IDT + CoIC (**Ours**) | ✓ | **4.6080**$_{\downarrow 0.0228}$ |

Table 5: Ablation on the proposed CoI-M modulation strategy, the contrastive learning loss, and the type of negative exemplars (rain- and detail-aware). Values marked with ↑ demonstrate the PSNR improvement against the first row.

| CoI-M | $L_{contra}$ | Negative exemplars | Rain200L | Rain200H | Rain800 |
|---|---|---|---|---|---|
| ✗ | ✗ | No | 40.26 | 31.42 | 31.10 |
| ✗ | ✗ | No | 40.27$_{\uparrow 0.01}$ | 31.46$_{\uparrow 0.04}$ | 31.16$_{\uparrow 0.06}$ |
| ✓ | ✓ | Rain-aware | 40.27$_{\uparrow 0.01}$ | 31.48$_{\uparrow 0.06}$ | 31.10 |
| ✓ | ✓ | Detail-aware | 40.27$_{\uparrow 0.01}$ | 31.48$_{\uparrow 0.06}$ | 31.20$_{\uparrow 0.10}$ |
| ✓ | ✓ | Rain-/Detail-aware | 40.27$_{\uparrow 0.01}$ | 31.46$_{\uparrow 0.04}$ | 31.33$_{\uparrow 0.23}$ |

**Real-world Deraining Transferred from Synthetic Datasets**. Our subsequent analysis examines the real-world deraining capabilities by training models *on an individual dataset* (*e.g.*, Rain800), *directly on mixed multiple datasets*, and *on mixed multiple datasets employing the proposed CoIC*. We select DGUNet and IDT as baselines. Additionally, we include the official pre-trained Restormer[4], denoted as Restormer[†], for comparison. Table 4 reports the evaluation comparisons. Both DGUNet and IDT trained solely on the Rain800 dataset demonstrates the worst performance. Training on the mixed multiple datasets has improved generalization for both DGUNet and IDT. Notably, IDT surpasses Restormer[†] when trained on mixed datasets. Equipped with the proposed CoIC, DGUNet and IDT achieves the best real-world deraining quality, validating the superiority of our proposed method. Visual results are provided in Appendix A.8.

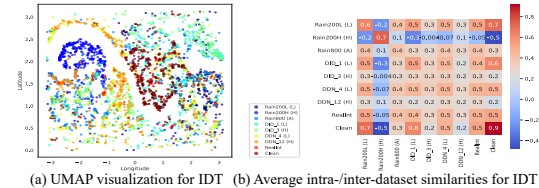

(a) UMAP visualization for IDT  (b) Average intra-/inter-dataset similarities for IDT

Figure 5: (a) UMAP visualization of learned representations. (b) Average intra-/inter-dataset similarities. "L", "H", and "A" indicate light, heavy, and accumulated rain, respectively.

**Further Training Incorporating Real-world Dataset**. To fully explore the potential of the proposed CoIC, we add a real-world dataset SPAData (Wang et al., 2019) to further train DRSformer

---

[4]https://github.com/swz30/Restormer

obtained using mixed synthetic datasets. The SPAData contains 28,500/1000 image pairs for training and evaluation. Surprisingly, we observe that DRSformer with CoIC has obtained remarkable real-world deraining ability as shown in Figure 4 (see more high-quality results in Appendix A.9). On the other hand, DRSformer with CoIC has significantly outperformed the direct training edition, as can be seen in Table 2. *This suggests that CoIC can help learn a comprehensive deraining model adept at handling both synthetic and real rains.* We also conduct comparison by training DRSformer on five synthetic datasets and SPAData *from scratch* in Appendix A.10.

**Joint Rain-/Detail-aware Representation Visualization**. We employ UMAP (McInnes et al., 2018) to project all instance-level representations onto a two-dimensional spherical surface. For simplicity, the *longitude* and *latitude* are plotted in Figure 5 (a). The IDT trained on mixed five synthetic datasets is selected for visualization, with 400 examples randomly sampled from each dataset: Rain200L, Rain200H, Rain800, DID_1, DID_3, DDN_4, and DDN_12 (details of datasets are in Appendix A.6). Additionally, all 146 examples from RealInt and 400 clean images in Rain200L are included. The results in Figure 5 (a) demonstrate that rainy images from different datasets may share similar embeddings. We further plot the intra-/inter-dataset similarities in Figure 5 (b) using these embeddings. More in-depth analyses are provided in the Appendix A.6.

## 4.3 ABLATION STUDY

**Effectiveness of the CoI-M**. We first study the efficacy of the proposed CoI-M. we train DGUNet on the mixed dataset of Rain200L, Rain200H, and Rain800. Quantitative results are tabulated in Table 5. Typically, DGUNet with CoI-M has improved PSNR by 0.06dB on the Rain800 dataset, indicating performance gains from learning adaptive deraining. Furthermore, a real-world comparison presented in Figure 6 suggest that DGUNet without CoI-M tends to overlook the real rain (Figure 6 (a)). In contrast, DGUNet with CoI-M readily perceives the real rain and efficiently eliminates the rain streaks (Figure 6 (b)).

**Effectiveness of the Rain-/Detail-aware Negative Exemplars**. We train DGUNet on the mixed dataset of Rain200L, Rain200H, and Rain800 to analyze the impact of rain-/detail-aware negative exemplars. Quantitative results are presented in Table 5. Incorporating detail-aware and rain-aware negative exemplars consecutively enhances the deraining performances. With both exemplar types, CoIC provides considerable PSNR gain of 0.23dB on the Rain800 dataset comprising complex rain. However, rain-aware negative exemplars alone brings no improvement on the Rain800 dataset. This occurs because the encoder finds a trivial solution of $\ell_{contra}$ (see in equation 4) by only discriminating heavy rain from other rain, owing to the $\arg\max$ in equation 3. Figure 6 (c) & (d) verify this explanation by visualizing the embedding space. In summary, rain-/detail-aware negative exemplars mutually reinforce the learning of instance-level representations, hence significantly improving performance.

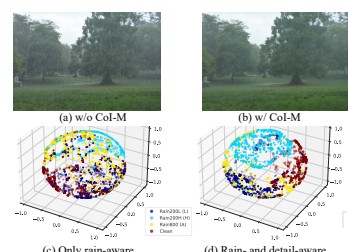

Figure 6: (a) & (b) A real-world deraining result w/o and w/ CoI-M. (c) & (d) UMAP visualization of learned embedding space.

## 5 CONCLUSION AND FUTURE WORK

Large image deraining models have sprung up recently pursuing superior performance. However, the potential of datasets remains nearly untapped. We find that inherent discrepancies among datasets may sub-optimize models, and shrinks their capabilities. To address this, we develop a novel and effective CoIC method for adaptive deraining. We first propose a novel contrastive learning strategy to explore joint rain-/detail-aware representations. Leveraging these representations as instructive guidance, we introduce CoI-M to perform layer-wise modulation of CNNs and Transformers. Experimental results on synthetic and real-world datasets demonstrate the superiority of CoIC. Moreover, CoIC enables exploring relationships across datasets, and model's behaviors. Furthermore, superior real-world deraining performances with CoIC are observed when further incorporating real-world dataset to train model. In the future, we anticipate extending CoIC to learn more practical deraining models that could handle diverse rains coupled with fog, dim light, blur, noise, and color shift. Moreover, expanding CoIC to all-in-one image restoration task is also promising.

## 6 ACKNOWLEDGEMENT

This work was supported by Key Area Support Plan of Guangdong Province for Jihua Laboratory (X190051TB190).

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

# A  APPENDIX

## A.1  DERIVATION OF THE CHANNEL-WISE TEMPERATURE

Please recall that the proposed CoI-M generates an adaptive weight $\mathbf{A}^l_{c,c',\alpha,\beta}$ following equation 8, which is re-written below:

$$\mathbf{A}^l_{c,c',\alpha,\beta} = \frac{k^2 e^{Z^l_{\alpha,\beta}/\tau^l_{cc'}}}{\sum_{\alpha',\beta'} e^{Z^l_{\alpha',\beta'}/\tau^l_{cc'}}}. \tag{12}$$

For a Softmax operation on any give score $\mathbf{s} = [s_1, s_2, \cdots, s_n]$ and a positive scalar temperature $t$, it outputs

$$\text{Softmax}(\mathbf{s}, t)_i = \frac{s_i/t}{\sum_j e^{s_j/t}}. \tag{13}$$

Generally, the score $\mathbf{s}$ is obtained by a shared classification head for all samples. However, the proposed CoI-M equips different convolutional layers with different MLPs that output both $Z^l_{\alpha,\beta}$ and $\tau^l_{cc'}$, hence the amplitude of $Z^l_{\alpha,\beta}$ in equation 12 also influences the Softmax result. To compare the generated temperatures for all convolutional layers, a proper normalization of $Z^l_{\alpha,\beta}$ is indispensable. Specifically, let $\bar{Z}^l_{\alpha,\beta} = Z^l_{\alpha,\beta} - \min Z^l_{\alpha,\beta}$, where $\bar{Z}^l_{\alpha,\beta} \geq 0$. Additionally, denote $\bar{Z}^l_{\alpha,\beta} = \mu^l \Gamma^l_{\alpha,\beta}$, where $\sum_{\alpha,\beta} \Gamma^l_{\alpha,\beta} = 1$. We can re-formulate equation 12 following equation 13 to

$$\mathbf{A}^l_{c,c',\alpha,\beta} = \frac{k^2 e^{\mu^l \Gamma^l_{\alpha,\beta}/\tau^l_{cc'}}}{\sum_{\alpha',\beta'} e^{\mu^l \Gamma^l_{\alpha',\beta'}/\tau^l_{cc'}}} = k^2 \text{Softmax}(\Gamma^l, \tau_{cc'}/\mu^l), \tag{14}$$

where $\Gamma^l$ is a normalization of $Z^l$ constrained by $\sum_{\alpha,\beta} \Gamma^l_{\alpha,\beta} = 1$. Note that $\mu^l$ can be calculated by

$$\mu^l = \frac{\sum_{\alpha,\beta} \bar{Z}^l_{\alpha,\beta}}{\sum_{\alpha,\beta} \Gamma^l_{\alpha,\beta}} = \sum_{\alpha,\beta} \bar{Z}^l_{\alpha,\beta} = \sum_{\alpha,\beta} (Z^l_{\alpha,\beta} - \min Z^l_{\alpha,\beta}). \tag{15}$$

Therefore the induced temperature is $\tau_{cc'} / \sum_{\alpha,\beta} (Z^l_{\alpha,\beta} - \min Z^l_{\alpha,\beta})$. Equivalently, we choose the temperature to be the form of $T^l_{cc'} = k^2 \tau_{cc'} / \sum_{\alpha,\beta} (Z^l_{\alpha,\beta} - \min Z^l_{\alpha,\beta})$, where we let the summation operation $\sum_{\alpha,\beta}$ to be an averaging operation. In this way, equation 14 becomes

$$\mathbf{A}^l_{c,c',\alpha,\beta} = k^2 \text{Softmax}(k^2 \Gamma^l, k^2 \tau_{cc'}/\mu^l) = k^2 \text{Softmax}(k^2 \Gamma^l, T^l_{cc'}) \tag{16}$$

**How does CoIC Modulate the Model**?  Formally, a large temperature $T^l_{cc'}$ produces an approximately uniform distribution of Softmax operation.  According to equation 16, all elements in $\mathbf{A}^l_{c,c'}$ nearly become 1, demonstrating no modulation (*all spatial weights of a convolutional kernel are important*).  Conversely, when $T^l_{cc'}$ is small, the generated $\mathbf{A}^l_{c,c'}$ will collapse to spatial coordinates with large $\Gamma^l_{\alpha,\beta}$, demonstrating the convolution layer is modulated to focus on local regions with a shrinked receptive field (*spatial weights of a kernel focusing on informative neighbors are important*). A shrinked receptive field of a kernel (*i.e.*, small temperature $T^l_{cc'}$) is observed in the deep encoder layers where skip connection is employed, as well as the deep decoder layers which produce high quality restored images (Please refer to Figure 7). We conjecture that the non-modulated deep encoder layers exhibit large receptive field due to downsampling and stacked convolutional layers. However, the skip connection from encoder to decoder for reconstructing details restricts that the features passed from encoder should capture rich image details without blur, hence the encoder is required to focus on informative local

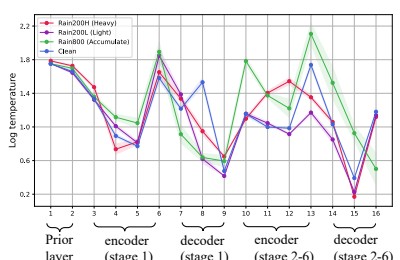

Figure 7: Average layer-wise $\log T^l_{cc'}$ with 95% confidence interval of DGUNet.

regions, corresponding to low temperatures. Conversely, the features passed to deep decoder layers contain rich details and possess high resolution, therefore small temperatures are required to focus on local regions to avoid blurry result.

**Algorithm 1** CoI-M for CNN layer, PyTorch-like

```
1  import torch.nn.functional as F
2  import torch.nn as nn
3  # Customize module to modulate CNN layers
4  class CoIM_Conv2d(nn.Module):
5      def __init__(self, in_c, out_c, k_size, stride, pad, grps, bias):
6          # grps: number of conv groups
7          self.bias = bias
8          self.stride = stride
9          self.padding = pad
10         self.groups = grps
11         self.conv = nn.Conv2d(in_c, out_c, k_size, stride,
12                      pad, groups=gps, bias=bias)
13     def forward(x, adaptive_w):
14         # x: shape (b, c, h, w)
15         # adaptive_w indicates generated weights using context in x and
       rain-/detail-aware representation z
16         if adaptive_w is None:
17             return self.conv(x)
18         else:
19             # adaptive_w: shape (b, out_c, in_c // grps, k_size, k_size)
20             b, c, h, w = x.shape
21             if self.bias:
22                 bias = self.conv.bias.repeat(b) # repeat bias
23             # modulate conv weight using adaptive_w
24             weight = self.conv.weight.unsqueeze(0) * adaptive_w # shape: (b,
       out_c, in_c // grps, k_size, k_size)
25             x = x.view(1, -1, h, w) # absorb batch_size dimension to channel
       dimension, shape (1, b*in_c, h, w)
26             weight = weight.view(-1, in_c // self.groups, k_size, k_size) #
       absorb batch_size to out_c dimension, shape: (b*out_c, in_c // grps,
       k_size, k_size)
27             out = F.conv2d(x, weight=weight, bias=bias, stride=self.stride,
28                       padding=self.padding, gropus=b*self.groups) # shape:
       (1, b*out_c, h', w')
29             return out.view(b, c, out.shape[2], out.shape[3])
```

To verify our conjecture, here we select the U-Net like DGUNet to elucidate how the receptive field is layer-wisely modulated given different inputs. Figure 7 plots layer-wise $\log T^l_{cc'}$ across the Rain200L, Rain200H, and Rain800 datasets. Deep encoder layers (with skip connections) and deep decoder layers exhibit small temperatures, concentrating more on informative local regions for detail restoration. Additionally, the encoder's bottleneck layer has large temperature, enabling rich deep feature extraction. Given different inputs, *differences manifest primarily in the decoder*, demonstrating that effective modulation occurs when features from encoders and decoders are interacted at different scales. These results have substantiated the above conjecture.

### A.2 PyTorch-like Pseudo Code for Equation 9

In Section 3.4, we propose a context-based instance-level modulation mechanism to modulate the weights of all CNN layers in an efficient way. Typically, we follow equation 9 which we re-write below:

$$\tilde{F}^{l+1} = (\mathbf{A}^l \mathbf{W}^l) \star F^l + b^l = F^{l+1} + \Delta \mathbf{W}^l \star F^l. \tag{17}$$

However, the generated adaptive weight $\mathbf{A}^l$ is instance-related, which requires a for-loop to calculate equation 17 for a data batch. The for-loop implementation will cause high computation cost and subsequently increase the training time. Therefore, we introduce Algorithm 1 to compute equation 17 in parallel with the help of group convolution operation.

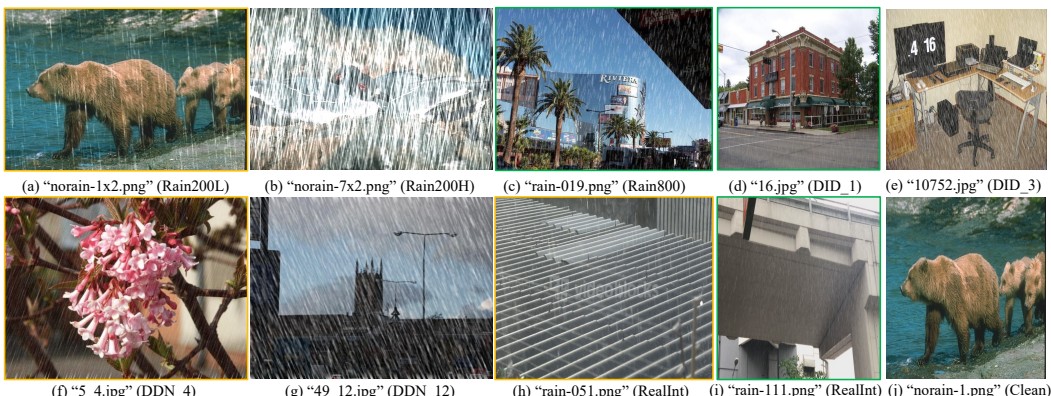

| (a) "norain-1x2.png" (Rain200L) | (b) "norain-7x2.png" (Rain200H) | (c) "rain-019.png" (Rain800) | (d) "16.jpg" (DID_1) | (e) "10752.jpg" (DID_3) |
| (f) "5_4.jpg" (DDN_4) | (g) "49_12.jpg" (DDN_12) | (h) "rain-051.png" (RealInt) | (i) "rain-111.png" (RealInt) | (j) "norain-1.png" (Clean) |

Figure 8: Visual examples of images from different datasets for instance-level representation visualization. Images marked with orange box contain similar *light* rain patterns. Images marked with green box are characterized by similar *accumulated* rain. Be aware that both (h) & (i) are real-world rainy images.

Table 6: Model complexity analysis for BRN, RCDNet, DGUNet, IDT, and DRSformer in terms of #P, FLOPs, and testing time. #P indicates the number of parameters. #$\Delta$P denotes the increased parameters with CoIC.

| Methods | Input size | #P (M) | #$\Delta$P w/ CoIC (M) | FLOPs (G) | FLOPs w/ CoIC (G) | Testing time (ms) | Testing time w/ CoIC (ms) |
|---|---|---|---|---|---|---|---|
| BRN | $512 \times 512$ | 0.38 | 0.21 | 392.9 | 393.3 | 333.2± 38.1 | 364.2± 33.0 |
| RCDNet | $512 \times 512$ | 2.98 | 2.11 | 389.0 | 389.5 | 351.6±0.2 | 384.8±2.2 |
| DGUNet | $512 \times 512$ | 3.63 | 1.62 | 396.8 | 397.2 | 161.0±5.5 | 198.8±6.8 |
| IDT | $128 \times 128$ | 16.42 | 2.53 | 7.3 | 7.6 | 59.8±2.7 | 83.2±10.5 |
| DRSformer | $512 \times 512$ | 33.67 | 14.12 | 440.8 | 441.1 | 833.6±0.5 | 886.7±11.0 |

## A.3 Model Complexity Analysis

In this section, we provide a comprehensive model complexity comparison between selected BRN, RCDNet, DGUNet, RCDNet, and DRSformer baselines and them equipped with the proposed CoIC. We evaluate the complexities in terms of the number of parameters (#P), FLOPs, and testing time. The results are provided in Table 6. Note that IDT can only accept input of size $128 \times 128$ due to spatial window-based self-attention mechanism. The increased parameters brought by CoIC depend on the intrinsic model architectures. By comparing the performance in Table 1 for IDT, we can conclude that the performance gain is not from the increased parameters. Additionally, the increased testing time comes mainly from the encoding process (about 30 ms), which demonstrates that the modulation process brings nearly no extra inference burden.

## A.4 Balance between Data-Fidelity Loss and Contrastive Loss

The hyperparameter $\lambda$ balances the contrastive loss contribution in equation 1. We train DGUNet on the mixed dataset of Rain200L, Rain200H, and Rain800 with $\lambda = [0.0, 0.05, 0.1, 0.2, 0.4, 0.8]$ to examine its impact. Figure 9 displays the results. Notably, the contrastive loss slight improves the performance on Rain200H and Rain200L, which are characterized by homogenous rain. In contrast, CoIC with contrastive loss stably improves Rain800 performance, demonstrates its superiority for complex and accumulated rain. We heuristically choose the best $\lambda = 0.2$ for our experiments.

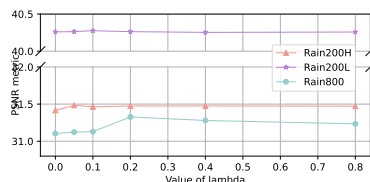

Figure 9: Ablation on the hyperparameter $\lambda$.

Table 7: Quantitative results of DRSformer trained on mixed synthetic datasets with different patch sizes.

| Methods | Patch size | Rain200L | | Rain200H | | Rain800 | | DID-Data | | DDN-Data | |
|---|---|---|---|---|---|---|---|---|---|---|---|
| | | PSNR ↑ | SSIM ↑ | PSNR ↑ | SSIM ↑ | PSNR ↑ | SSIM ↑ | PSNR ↑ | SSIM ↑ | PSNR ↑ | SSIM ↑ |
| DRSformer | $96 \times 96$ | 39.74 | 0.9858 | 30.42 | 0.9057 | 29.86 | 0.9114 | 34.96 | 0.9607 | 33.92 | 0.9541 |
| DRSformer + CoIC (**Ours**) | $96 \times 96$ | **39.81** | **0.9862** | **30.50** | **0.9076** | **29.92** | **0.9115** | **35.01** | **0.9614** | **33.94** | **0.9545** |
| DRSformer | $128 \times 128$ | **39.97** | 0.9868 | 30.78 | 0.9112 | 30.09 | 0.9114 | 35.13 | 0.9624 | 34.03 | 0.9556 |
| DRSformer + CoIC (**Ours**) | $128 \times 128$ | 39.95 | **0.9869** | **30.83** | **0.9118** | **30.21** | **0.9159** | **35.17** | **0.9629** | **34.11** | **0.9567** |

## A.5 TRAINING DETAILS AND HISTORIES OF ALL MODELS

In this section, we provide training details for BRN, RCDNet, DGUNet, IDT, and DRSformer. Specifically, both BRN w/o and w/ CoIC is trained on $100 \times 100$ image patches, with a batch size of 12 for about 260k iterations. We train RCDNet w/o and w/ CoIC on image patches of size $64 \times 64$ with batch size 16 for about 260k iterations. For the large DGUNet, we train it w/o and w/ CoIC on image patches of size $128 \times 128$ with batch size 16 for about 400k iterations until converge. The Transformer model IDT w/o and w/ CoIC are trained on image patches of size $128 \times 128$ with batch size 8 for about 300k iterations. We train DRSformer on mixed synthetic datasets on $96 \times 96$ image patches with batch size 4. A $96 \times 96$ image patch setting enables DRSformer to fit on a single NVIDIA 3090 24G GPU. However, since DRSformer utilizes spatial sparse attention mechanism, a small image patch size may degrade the performances. Therefore, we next train DRSformer w/o and w/ CoIC on $128 \times 128$ image patches, which further improves the performances on five synthetic datasets (Please refer to Table 7).

For the purpose of further training DRSformer pre-trained on mixed synthetic datasets with real-world SPAData. We continue to train both DRSformer w/o and w/ CoIC on mixed synthetic and SPAData for about another 105k iterations. Figure 10 (a)-(e) display the training data-fidelity loss curves for BRN, RCDNet, DGUNet, IDT, and DRSformer trained on mixed synthetic datasets. Figure 10 (f) shows the further training loss history of DRSformer after incorporating real-world SPA-Data dataset. The comparison in Figure 10 (f) demonstrates that the proposed CoIC can help DRSformer learn better when exposed to both synthetic and real-world data.

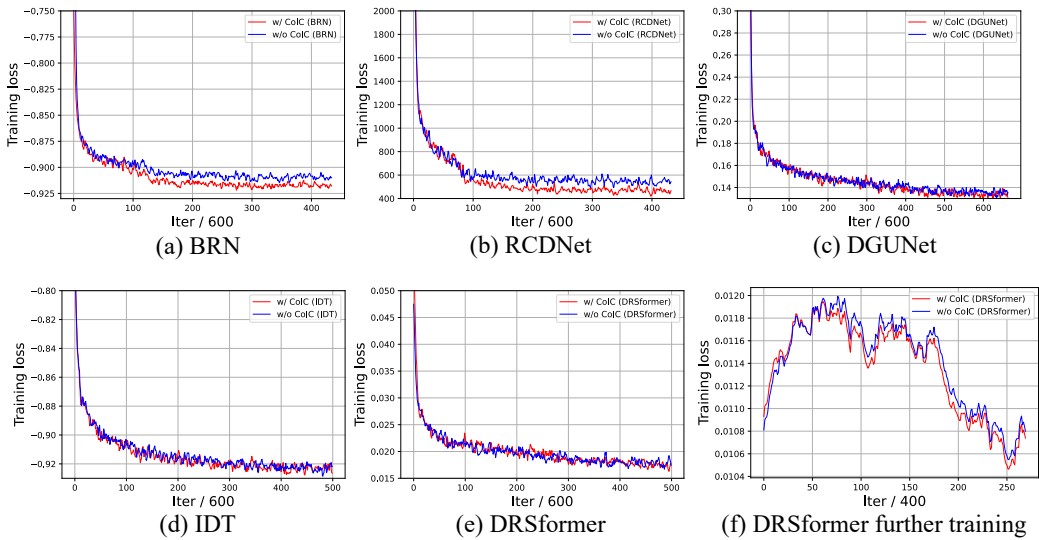

Figure 10: Training loss curves for BRN, RCDNet, DGUNet, IDT, and DRSformer.

## A.6 ANALYSIS ON JOINT RAIN-/DETAIL-AWARE REPRESENTATIONS

We present the dataset details for the visualization of instance-level representations in Section 4.2. As stated in Section 4.2, 400 examples from Rain200L, Rain200H, Rain800, DID_1, DID_3, DDN_4,

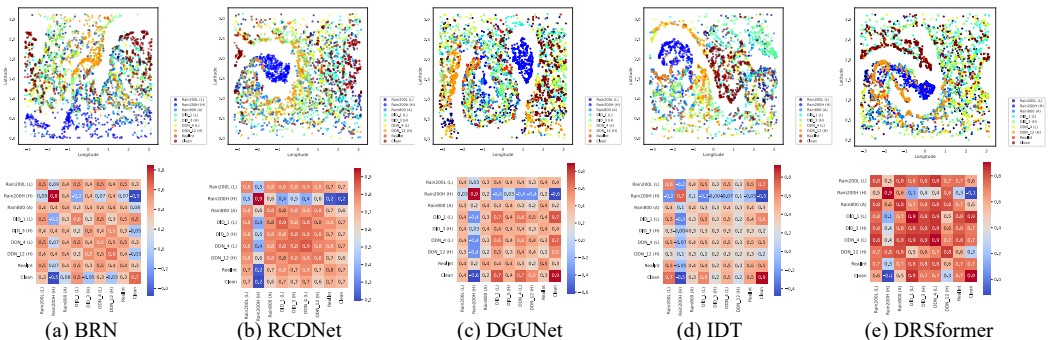

Figure 11: Top: Joint rain-/detail-aware representations visualization for different models. Bottom: Average intra- and inter-dataset similarities learned of different models utilizing the proposed CoIC. "L", "H", and "A" represent light, heavy, and accumulate rain, respectively.

DDN_12 are randomly sampled for visualization. Additionally, all 146 real-world rainy images from RealInt and 400 clean images (noted as Clean) from Rain200L are included. Recall that DID-Data comprises three density levels (*i.e.*, low, medium, and high), hence we denote the first (smallest) density level as DID_1, while the third (largest) density level as DID_3. In other words, the images in DID_1 and DID_3 are characterized with light rain and heavy rain, respectively. As for the DDN-Data which contains 14 kinds of rain augmentation, we select its fourth augmentation with light rain (DDN_4) and twelfth augmentation (DDN_12) with heavy rain for visualization. We also present a visual example for these datasets in Figure 8, where their rain patterns can be well perceived visually. We can easily observe that the real-world rainy image in Figure 8 (h) comprises *light* rain pattern similar to synthetic images in Figure 8 (a) (from Rain200L) and Figure 8 (f) (from DDN_4). Additionally, the real-world rainy image in Figure 8 (i) contains similar *accumulated* rain to synthetic rainy images in Figure 8 (c) (from Rain800) and Figure 8 (d) (from DID_1). These observations suggest that *training models on mixed multiple datasets can improve its generalization ability on diverse real-world rainy images*.

With images from Rain200L, Rain200H, Rain800, DID_1, DID_3, DDN_4, DDN_12, RealInt, and Clean, we can employ the pre-trained feature extractor to obtain their instance-level representations. We project these high-dimensional representations onto a two-dimensional spherical surface. For better visualization, we plot the *longitude* and *latitude* of these projected representations. Further, these representations enable us to calculate the intra- and inter-dataset average similarities, which in turn reveal the relationships among datasets. Specifically, BRN, RCDNet, DGUNet, IDT, and DRSformer trained on mixed five synthetic datasets (Rain200L & H, Rain800, DID-Data, and DDN-Data) utilizing the proposed CoIC are selected. The visualization results are displayed in Figure 11 for all these models. It can be seen that the learned embedding space of all five models are *neither density-level nor dataset-level discriminative*. This is due to the fact that rainy images from two different datasets may contain similar rain pattern (see Figure 8 (a) *vs.* Figure 8 (f), and Figure 8 (c) *vs.* Figure 8 (e)). Therefore, the *rain-aware property of instance-level representations may not contribute to dataset-level discriminative embedding space*. On the other hand, the ground truths of different datasets may come from the same dataset, *e.g.*, BSD (Martin et al., 2001) and UCID (Schaefer & Stich, 2003) dataset, hence the *detail-aware property of instance-level representations may not result in well-separated dataset-level or density-level embedding space*.

The learned average intra-/inter-dataset similarities are provided in Figure 11. The average intra-/inter-dataset similarities learned by different models varies, demonstrating that a pre-defined criterion to discriminate rain and details cannot be applied to all models. For instance, both as transformers, IDT tends to be more sensitive to the differences between light and heavy rain than DRSformer as shown in Figure 11 (d) & (e). Furthermore, the negative similarity between two datasets (*e.g.*, Rain200L and Rain200H for IDT in Figure 11 (d)) may imply *dataset collision* or *dataset competition* when directly training models on these datasets. Our proposed CoIC method encourages different models to learn decent joint rain-/detail-aware embedding spaces.

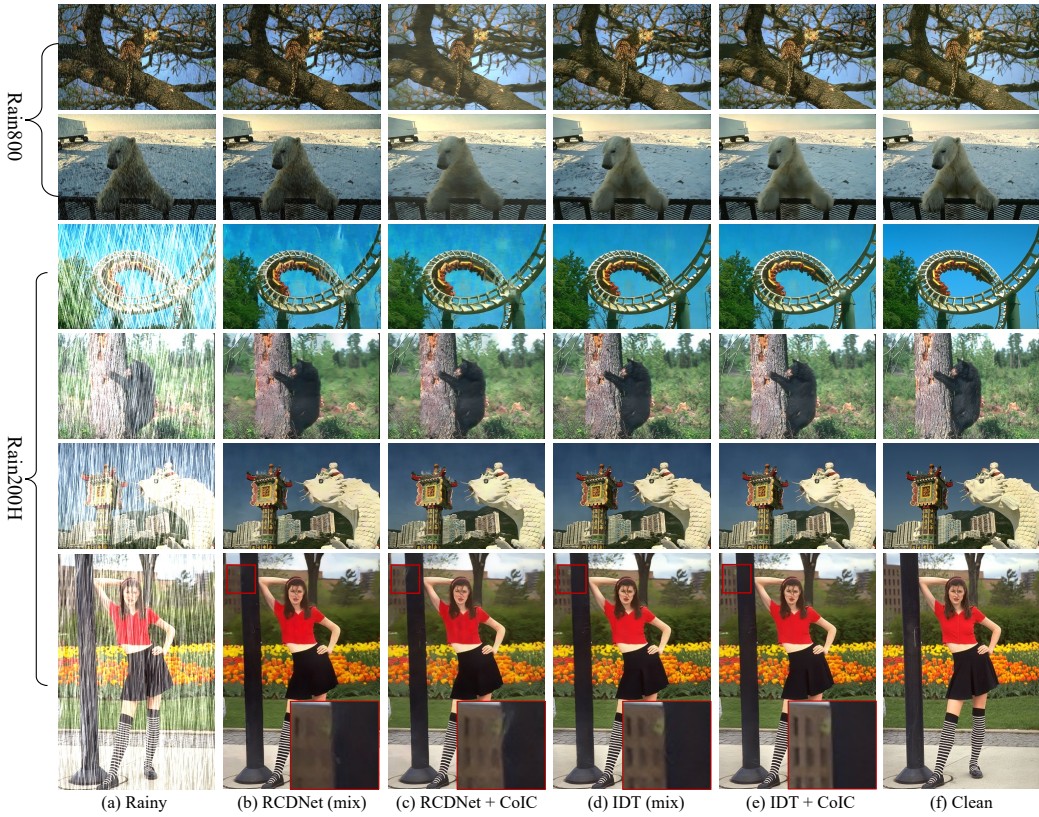

Figure 12: Qualitative comparison for RCDNet and IDT. Please zoom in for more details.

We conduct a visual deraining comparison on synthetic datasets in this section to demonstrate the effectiveness of the proposed CoIC method. Concretely, two examples from Rain800 dataset and four rainy images from the heavy rain Rain200H dataset are selected. We utilize pre-trained RCDNet and IDT on mixed five synthetic datasets for qualitative evaluation. Figure 12 displays the result. It can be seen from Figure 12 that both RCDNet and IDT employing the proposed CoIC can better eliminate rain effect as well as recover details, owing to the joint rain-/detail-aware guidance.

## A.8 MORE REAL-WORLD DERAINING RESULTS BY USING SYNTHETIC DATA

In this section, we provide more real-world deraining examples to examine the generalization abilities of models trained *on individual dataset* (*e.g.*, Rain800), *directly on mixed multiple datasets*, and *on mixed datasets employing the proposed CoIC*. Typically, we select the representative CNN model DGUNet, along with the Transformer model IDT as baseline models. Additionally, the official powerful pre-trained Restormer, noted as Restormer[†], is included for comparison. The qualitative results are presented in Figure 13, where images contaminated with various kinds of rain are included. Models (DGUNet and IDT) trained on an individual dataset tend to overlook rain streaks with high pixel intensities or result in over-smooth results, demonstrating poor perceptual abilities on rain and image details. In contrast, models trained directly on multiple datasets can well handle complex rain occasions. However, these models may fail to strike a good balance between removing rain and preserving details (see the first, third, and fourth row in Figure 13 (g)). On the contrary, models trained utilizing CoIC can well balance rain removal and detail restoration (see the first, third, and fourth row in Figure 13 (h)), owing to the exploration of joint rain-/detail-aware representations. These results have verified the superiority of the proposed CoIC method.

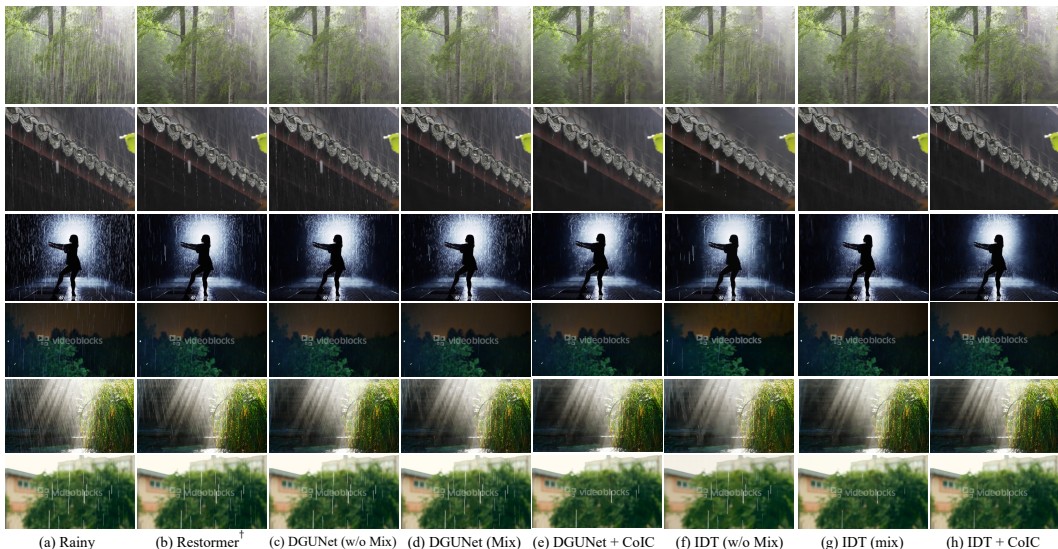

| (a) Rainy | (b) Restormer† | (c) DGUNet (w/o Mix) | (d) DGUNet (Mix) | (e) DGUNet + CoIC | (f) IDT (w/o Mix) | (g) IDT (mix) | (h) IDT + CoIC |

Figure 13: More real-world deraining comparison results for DGUNet and IDT. Please zoom in for more details.

Table 8: Quantitative results of DRSformer w/o and w/ CoIC trained on mixed synthetic and real-world datasets.

| Methods | Training | Rain200L | | Rain200H | | Rain800 | | DID-Data | | DDN-Data | | SPAData | |
|---|---|---|---|---|---|---|---|---|---|---|---|---|---|
| | | PSNR ↑ | SSIM ↑ | PSNR ↑ | SSIM ↑ | PSNR ↑ | SSIM ↑ | PSNR ↑ | SSIM ↑ | PSNR ↑ | SSIM ↑ | PSNR ↑ | SSIM ↑ |
| DRSformer | *two-stage* | 39.32 | 0.9850 | 29.27 | 0.9000 | 28.85 | 0.9070 | 34.91 | 0.9607 | 33.71 | 0.9540 | 45.46 | 0.9898 |
| DRSformer + CoIC (**Ours**) | | **39.70** | **0.9860** | **30.31** | **0.9058** | **29.73** | **0.9143** | **35.02** | **0.9618** | **33.94** | **0.9556** | **46.03** | **0.9903** |
| DRSformer | *from scratch* | 39.39 | 0.9850 | 29.64 | 0.8948 | 28.88 | 0.9058 | 34.83 | 0.9594 | 33.66 | 0.9521 | 46.52 | 0.9899 |
| DRSformer + CoIC (**Ours**) | | **39.53** | **0.9854** | **29.88** | **0.8959** | **29.27** | **0.9069** | **34.84** | **0.9597** | **33.79** | **0.9526** | **46.76** | **0.9907** |

## A.9 MORE REAL-WORLD VISUAL DERAINING RESULTS BY TRAINING INCORPORATING SPADATA

Here we provided more visual real-world deraining results by further training DRSformer w/o and w/ the proposed CoIC incorporating real-world SPAData dataset. The results are displayed in Figure 14, where we can observe that DRSformer w/ CoIC can achieve much better deraining performances.

## A.10 TRAINING ON SYNTHETIC AND REAL-WORLD DATASETS FROM SCRATCH

In Section 4.2, we have explored the potential of the proposed CoIC by incorporating a real-world dataset SPAData and further training DRSformer. This *two-stage training* approach allows the pre-trained DRSformer to first assimilate deraining knowledge from five synthetic datasets, thereby facilitating its learning process upon the addition of real-world SPAData. Therefore, we further conduct experiments on training DRSformer utilizing both five synthetic datasets and real-world SPA-Data *from scratch*, where the larger discrepancies among datasets may impede the learning process. In practice, we train DRSformer w/o and w/ the proposed CoIC for about 405K iterations, which equals to the total training iterations of the two-stage approach. Quantitative results are provided in Table 8, where DRSformer with the proposed CoIC has outperformed vanilla DRSformer overall in six datasets, demonstrating the effectiveness of the devised method. Surprisingly, compared to the two-stage training approach, DRSformer trained from scratch has not shown superior performance among Rain200L, Rain200H, Rain800, DID-Data, and DDN-Data. Consequently, the improvement brought by the proposed CoIC shrinks. However, DRSformer training from scratch has exhibited remarkable deraining ability on the real-world dataset SPAData, achieving 46.52dB (w/o CoIC) and 46.76dB (w/ CoIC) in terms of PSNR metric. We conjecture that this counter-intuitive result is due to the extremely imbalanced data scale and small batch size. The official SPAData contains 28,500 training image pairs with high resolution, resulting in a training dataset containing 638,473 examples. When training DRSformer using five synthetic datasets and SPAData from scratch with

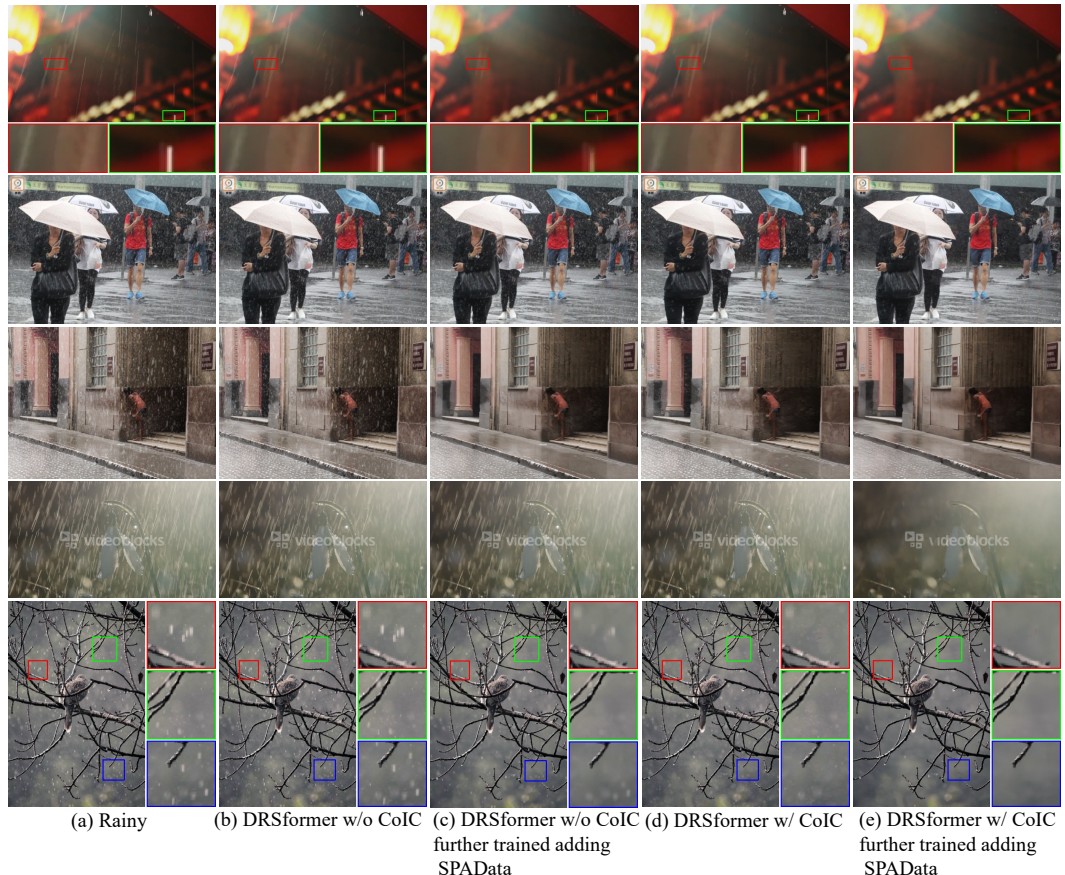

| (a) Rainy | (b) DRSformer w/o CoIC | (c) DRSformer w/o CoIC further trained adding SPAData | (d) DRSformer w/ CoIC | (e) DRSformer w/ CoIC further trained adding SPAData |

Figure 14: More real-world deraining comparison results by further training DRSformer w/o and w/ CoIC after adding SPAData. Please zoom in for more details.

a batch size of 4, almost all images are from SPAData throughout the training process, impeding the proposed contrastive learning process and further diminishing the performance gain on the five synthetic datasets. Conversely, as the training is primarily dominated by SPAData, both DRSformer without and with CoIC have achieved remarkable PSNR and SSIM metrics. In summary, *extreme data scale imbalance accompanying small batch size* is harmful and should be circumvented during the training process.

