# OpenReview forum: "Harnessing Joint Rain-/Detail-aware Representations to Eliminate Intricate Rains"
_ICLR.cc/2024/Conference — ICLR 2024 poster_

### Official Review · Reviewer_rX1q · 2023-10-17

**Soundness:** 3 good
**Presentation:** 3 good
**Contribution:** 3 good
**Rating:** 8
**Confidence:** 4

**Summary:**

Summary.

This paper proposes the CoIC algorithm to train CNN and Transformer based models to be instance-specific, using mixed datasets, for single image rain removal. The CoIC algorithm includes a Context-based Instance-specific Modulation (CoI-M) mechanism and a rain-/detail-aware contrastive learning strategy.

**Strengths:**

Strengths.

The paper is generally easy to read and follow.
The proposed approach seems to work well for CNN based models.
Experiments (with tables and figures) are provided to analyze the proposed method.
The proposed method seems to work well on the RealInt (Wang et al., 2019), in terms of NIQE.
The idea of generating negative examples for both rain and background and modulating the image-specific information into networks sounds interesting.

**Weaknesses:**

Weaknesses/Questions.


The Introduction mentions that there is a long-tail distribution across mixed datasets. As most (if not all) datasets are synthetic, would it be easier/better to synthesize more rain images to handle such an unbalanced distribution? Another question is whether training using a mixed dataset improves the individual performance of one model on each dataset. These answers are not easily found in this paper.

Figure 1(b) shows one observation of this paper that the image is gradually dominated by rain when rain density increases, which, however, sounds superficial. The resulting (from this observation) learning of a joint embedding space for both rain and detail information is based on contrastive learning, while it is not explained why using contrastive learning here as it seems that rain density is the most important factor. Meanwhile, the paper simply says it is distinct from previous instance-discriminative contrastive learning (He et al., 2020) without explanation.

The Introduction mentions that the embedding space of this paper is not instance-level discriminative, which is hard to understand as the paper aims to learn instance-specific deraining features (If not discriminative enough, how can one model learn the specific information?) The second point it mentions is that the background content is not discriminative, which I do not know whether it is correct, does not explain why it is not suitable for a standard contrastive learning strategy. After I read the method section, it seems to me that the main difference is to construct negative examples based on rain types (selected from another dataset) and Gaussian blurred background. The authors may correct me if I misunderstood but it seems to me that the discussion in the introduction does not match what they do in the method.

The paper uses Gaussian blur to construct negative exemplars of the background image, which needs further justification as it may not represent the degraded backgrounds. Usually when rain streaks are large or dense, there are more occlusions than blur, which the Gaussian blur may not model.



The related work section should be expanded. 1. There are certainly several rain removal methods not cited, especially for heavy rain restoration, although they do not combine multiple datasets for training. 2. The modulation is one of the key techniques in the proposed method but there is no literature review on it.

In Table 1, the performance gain over DRSformer (Chen et al., 2023) is rather limited. The paper needs to show more advantages, otherwise, it seems not worth doing considering the efforts and the performance gains.

The visual results are not impressive, as we may easily see the remained rain streaks in, e.g., Figure 4, Figure 12. I understand that there are improvements by comparing the results of methods with and without CoIC. However, it seems to me that these visual results are still failure cases (due to obvious rain streaks, in Figure 12, (g) is comparable to or even better than (h)). Combining it with Table 1, really makes others wonder about the necessity of using the proposed approach.

%%%%%%%%%%%%%%%%%%%%%%%%%%%%%%%%%%%%%%%%%%%%%%%%%%
I am glad to see that the rebuttal addresses my previous concerns/questions. I would like to raise my rating to accept and I hope that the authors will incorporate all information in this rebuttal into their final revision.

**Questions:**

Please see above.

---

> ### Author Response · Authors · 2023-11-20
> **Response to Reviewer rX1q (Part 1/4)**
>
> **Comment**: We appreciate the effort and valuable questions from the reviewer. We are also glad that the reviewer finds it easy to read and follow our work. Below is our point-to-point response:
>
> > The Introduction mentions that there is a long-tail distribution across mixed datasets. As most (if not all) datasets are synthetic, would it be easier/better to synthesize more rain images to handle such an unbalanced distribution? Another question is whether training using a mixed dataset improves the individual performance of one model on each dataset. These answers are not easily found in this paper.
>
> **As it may be helpful to synthesize more rain images to handle the unbalanced distribution, it is important to consider the potential large domain gap between synthetic rainy images and real-world rainy images.** A key principle in synthesizing rain images is to align the distribution between synthetic and real-world rainy images, especially for real-life applications. Intuitively, real-world rainy images may follow a long-tailed distribution (extremely heavy rainy images are rare). In fact, many researchers are dedicated to synthesizing **realistic images** to enhance deraining ability [1, 2], while paying little attention to balancing the distribution. More importantly, as highlighted by the latest research [2], synthesizing heavy rainy images may also result in more artifacts, which can harm performance. **Therefore, we believe it could be more valuable to synthesize more realistic images to train models.**
>
> As for the second question, large CNN and Transformer models can easily overfit to an individual dataset, as also noted by the reviewer nGEY. Typically, synthetic training and testing datasets possess a similar rain type. Therefore, it is easy for a model trained on an individual dataset to achieve extremely high PSNR/SSIM metrics on the corresponding testing set. However, **due to heavy overfitting, these models may struggle to handle rainy images from another dataset, thus limiting their applications**. Our analysis in Table 4 verifies that both DGUNet and IDT, when trained on an individual dataset, exhibit the worst performances on RealInt. In this paper, our goal is to learn a comprehensive deraining model that excels at deraining diverse rainy images. Also, we anticipate that a model trained on a mixed dataset could demonstrate better performance than one trained on an individual dataset.
>
> [1]. Hong Wang et al. "From rain generation to rain removal." in CVPR 2021.
>
> [2]. Shen Zheng et al. "TPSeNCE: Towards Artifact-Free Realistic Rain Generation for Deraining and Object Detection in Rain." in WACV 2023.

---

> > ### Author Response · Authors · 2023-11-20
> > **Response to Reviewer rX1q (Part 2/4)**
> >
> > >Figure 1(b) shows one observation of this paper that the image is gradually dominated by rain when rain density increases, which, however, sounds superficial. The resulting (from this observation) learning of a joint embedding space for both rain and detail information is based on contrastive learning, while it is not explained why using contrastive learning here as it seems that rain density is the most important factor. Meanwhile, the paper simply says it is distinct from previous instance-discriminative contrastive learning (He et al., 2020) without explanation.
> >
> > We want to clarify that the observations in Figure 1(b) are non-trivial, which may sound superficial to some extent. In fact, **we proposed a measure to quantitatively illustrate the relationships between rain/detail cues and rain density.** Second, *although the rain factor becomes dominant when rain density increases, the number of heavy rainy images drops dramatically, as illustrated in Figure 1 (a).* Please note that rainy images with relatively small density account for a large proportion. Therefore, **to learn a model for diverse rainy images, both rain-aware and detail-aware information should be considered.** In contrast, most recent contrastive learning-based [1,2,3] focus on exploring degradation-aware information, where detail-aware information is overlooked. Therefore, our findings from Figure 1 (a) & (b) provide insight into considering joint rain-/detail-aware representations for image deraining. In practice, our ablation study on negative exemplar types also substantiates that the detail-aware information can enhance deraining performance on Rain800 (please refer to Table 5 for details). Additionally, **only considering rain-aware representations may cause a trivial solution for the embedding space, where only heavy rains are distinguishable** (please refer to Figure 6 (c) & (d) for details).
> >
> > Since almost all rain datasets only provide ground truth as labels, we can only learn the representations in an unsupervised manner. Therefore, considering the success of contrastive learning-based methods, we designed a novel contrastive learning strategy to learn joint rain-/detail-aware representations. **Note that the contrastive learning method in (He et al., 2020) [4] is only content-related discriminative, neglecting the rain factor; thus, it cannot be utilized to extract rain-aware representations.**
> >
> > [1]. Xiang Chen, et al. "Unpaired deep image deraining using dual contrastive learning." in CVPR 2022.
> >
> > [2]. Longguang Wang, et al. "Unsupervised degradation representation learning for blind super-resolution." in CVPR 2021.
> >
> > [3]. Boyun Li, et al. "All-in-one image restoration for unknown corruption." in CVPR 2022.
> >
> > [4]. Kaiming He, et al. "Momentum contrast for unsupervised visual representation learning." in CVPR 2020.

---

> > > ### Author Response · Authors · 2023-11-20
> > > **Response to Reviewer rX1q (Part 3/4)**
> > >
> > > > The Introduction mentions that the embedding space of this paper is not instance-level discriminative, which is hard to understand as the paper aims to learn instance-specific deraining features (If not discriminative enough, how can one model learn the specific information?) The second point it mentions is that the background content is not discriminative, which I do not know whether it is correct, does not explain why it is not suitable for a standard contrastive learning strategy. After I read the method section, it seems to me that the main difference is to construct negative examples based on rain types (selected from another dataset) and Gaussian blurred background. The authors may correct me if I misunderstood but it seems to me that the discussion in the introduction does not match what they do in the method.
> > >
> > > Thank you for pointing out this ambiguity, and we apologize for not clearly clarifying our motivation for employing contrastive learning, as well as the differences from other methods. We want to address the ambiguity caused by our original explanation (**We have revised this part in the *Introduction*, where we provide an in-depth discussion on existing contrastive learning methods, and we also remove all instance-specific phrases for better understanding**). The embedding space is not instance-level discriminative, meaning that **rainy images are not discriminative based on image content due to the coupled rain.** Therefore, the content-based contrastive learning approach [1] is rendered inapplicable. In fact, **our goal is to extract both rain- and detail-aware information that characterizes the rain component and background component. This rain-/detail-aware information, in turn, provides guidance to adaptively separate the rain from the rainy input.** Therefore, we need to learn an encoder that can well perceive different rains and background details in an unsupervised manner. Inspired by the success of unsupervised contrastive learning-based image restoration methods, we have designed a novel contrastive learning strategy that enables us to extract both rain- and detail-related representations.
> > >
> > > Existing contrastive learning methods mainly focus on extracting only the degradation factor [2,3,4] or aim at discriminating between rainy and clean images [5]. Therefore, these methods cannot effectively extract joint rain-/detail-related information from the rainy input. Moreover, the rain-aware negatives are not simply selected from another dataset, since *images from different datasets may share similar rain patterns*. Instead, **we maintain a *rain layer bank* during training, which is used to retrieve the most dissimilar rain pattern to construct negative exemplars.** Based on the proposed contrastive learning strategy, we can quantitatively assess the importance of rain and detail cues for deraining.
> > >
> > > To clearly underline the novelty of the proposed contrastive learning, we have revised the section on *Introduction* to better explain the limitations of existing approaches, as well as to clarify the differences from these approaches.
> > >
> > > > The paper uses Gaussian blur to construct negative exemplars of the background image, which needs further justification as it may not represent the degraded backgrounds. Usually when rain streaks are large or dense, there are more occlusions than blur, which the Gaussian blur may not model.
> > >
> > > Please note that Gaussian blur is used to **blur the details in the ground truth**, which serves as negative exemplars and should be pushed apart from the rainy image. When the rain streaks are large or dense, more occlusions than blur occur, which decreases the discriminability of details, as the reviewer pointed out. **But recall that when rain streaks are large or dense, the rain-aware information dominates; hence, the effect caused by the limitation of Gaussian blur can be neglected.** **With the help of the rain-aware negatives and detail-aware negatives, the proposed contrastive learning strategy can function well with either slight rain or dense rain.** In practice, we find that Gaussian blur works well.
> > >
> > > [1]. Kaiming He, et al. "Momentum contrast for unsupervised visual representation learning." in CVPR 2020.
> > >
> > > [2]. Xiang Chen, et al. "Unpaired deep image deraining using dual contrastive learning." In CVPR 2022.
> > >
> > > [3]. Longguang Wang, et al. "Unsupervised degradation representation learning for blind super-resolution." in CVPR 2021.
> > >
> > > [4]. Boyun Li, et al. "All-in-one image restoration for unknown corruption." in CVPR 2022.
> > >
> > > [5]. Yuntong Ye, et al. "Unsupervised deraining: Where contrastive learning meets self-similarity." in CVPR 2022.

---

> > > > ### Author Response · Authors · 2023-11-20
> > > > **Response to Reviewer rX1q (Part 4/4)**
> > > >
> > > > > The related work section should be expanded. 1. There are certainly several rain removal methods not cited, especially for heavy rain restoration, although they do not combine multiple datasets for training. 2. The modulation is one of the key techniques in the proposed method but there is no literature review on it.
> > > >
> > > > Thanks for your constructive advice. We have added four methods: a heavy rain removal method [1], two semi-supervised methods Syn2Real [2] and MOSS [3], and one unsupervised method NLCL [4]. Additionally, we have included a literature review of modulation techniques in image restoration entitled "Image Restoration with Modulation" in the *Related Work section.*
> > > >
> > > > > In Table 1, the performance gain over DRSformer (Chen et al., 2023) is rather limited. The paper needs to show more advantages, otherwise, it seems not worth doing considering the efforts and the performance gains. The visual results are not impressive, as we may easily see the remained rain streaks in, e.g., Figure 4, Figure 12. I understand that there are improvements by comparing the results of methods with and without CoIC. However, it seems to me that these visual results are still failure cases (due to obvious rain streaks, in Figure 12, (g) is comparable to or even better than (h)). Combining it with Table 1, really makes others wonder about the necessity of using the proposed approach.
> > > >
> > > > We acknowledge that the performance gain over DRSformer is marginal, mainly due to the Mixture of Experts (MoEs) in it. **However, we may have underestimated the potential of the proposed CoIC.** Considering suggestions from other reviewers, we have added a real-world dataset SPAData [5] to tune DRSformer for about 105k iterations. We tabulate the quantitative results below (please see details in Table 2):
> > > >
> > > > | Methods            | Rain200L  | Rain200H  | Rain800   | DID-Data  | DDN-Data  | SPAData   |
> > > > | ------------------ | --------- | --------- | --------- | --------- | --------- | --------- |
> > > > | DRSformer w/o CoIC | 39.32     | 29.27     | 28.85     | 34.91     | 33.71     | 45.46     |
> > > > | DRSformer w/ CoIC  | **39.70** | **30.31** | **29.73** | **35.02** | **33.94** | **46.03** |
> > > >
> > > > Note that **the proposed method has achieved a 1.04 dB, 0.88 dB, and 0.57 dB improvement on Rain200H, Rain800, and SPAData.** Obviously, the MoEs cannot handle the mixed synthetic and real-world rains effectively. In contrast, the proposed CoIC enables us to obtain a comprehensive deraining model adept at deraining diverse synthetic and real-world rains. Furthermore, we have also observed a significant deraining ability boost on RealInt. Please see our anonymous GitHub page: https://anonymous.4open.science/r/CoIC-730F/
> > > >
> > > > With the success after tuning DRSformer with CoIC on SPAData, we provide high-quality real-world deraining results in Figure 4 and Figure 14. For convenience, please review it on our anonymous GitHub page: https://anonymous.4open.science/r/CoIC-730F/
> > > >
> > > > In summary, the results of tuning on SPAData can well substantiate the effectiveness of the proposed CoIC. **In the future, we plan to extend the proposed method for image deraining under diverse rains coupled with fog, dim light, blur, and noise.**
> > > >
> > > > [1]. Ruoteng Li et al. "Heavy rain image restoration: Integrating physics model and conditional adversarial learning." in CVPR 2019.
> > > >
> > > > [2]. Yasarla, Rajeev, et al. "Syn2real transfer learning for image deraining using gaussian processes." in CVPR 2020.
> > > >
> > > > [3]. Huaibo Huang, et al. "Memory oriented transfer learning for semi-supervised image deraining." in CVPR 2021.
> > > >
> > > > [4]. Yuntong Ye, et al. "Unsupervised deraining: Where contrastive learning meets self-similarity." in CVPR 2022.
> > > >
> > > > [5]. Tianyu Wang et al. "Spatial attentive single-image deraining with a high quality real rain dataset." in CVPR 2019.

---

### Official Review · Reviewer_iZXv · 2023-10-31

**Soundness:** 2 fair
**Presentation:** 2 fair
**Contribution:** 2 fair
**Rating:** 6
**Confidence:** 5

**Summary:**

Existing de-raining methods tend to overlook the inherent differences between datasets. To address this limitation, this paper develops a rain-/detail-aware contrastive learning strategy to extract a representation, and proposes a Context-based Instance-specific Modulation mechanism, which uses the representation to modulate models. This approach helps existing methods boost the de-raining ability.

**Strengths:**

1. This paper focuses on training de-raining models on amalgamated datasets.
2. The experiments are sufficient and the results are better than others.
3. Visualization results on RealInt demonstrate the generalization capability of this method on real scenes.

**Weaknesses:**

1. The main concern is the necessity of the proposed method. In fact, when the deraining model is strong enough, it may be able to learn the differences between datasets. For example, in Table 1, the proposed method has limited performance improvement on DRSformer and DGUNet.
2. The novelty of the method may be limited. The method of representation extraction based on contrastive learning has been widely explored in the blind super-resolution field [1]. The paper needs to further explain the differences with them.
[1] Unsupervised Degradation Representation Learning for Blind Super-Resolution. CVPR 2021.
3. It would be better to show the increase in the number of parameters and changes in inference time brought by the proposed method.

**Questions:**

Please see 'Weaknesses'.

---

> ### Author Response · Authors · 2023-11-20
> **Response to Reviewer iZXv (Part 1/2)**
>
> Comments: We thank the reviewer for the insightful questions, and we hope to address the concerns with our response below.
>
> > Limited performance improvement in Table 1
>
> The performance improvement in Table 1 for DGUNet and DRSformer seems marginal, primarily because these models may be capable enough to process synthetic datasets. For instance, DRSformer contains a Mixture of Experts (MoEs) that enables it to tolerate different rains. **However, the potential of the proposed CoIC may be underestimated only based on the results from synthetic datasets.** As suggested by other reviewers, we have added a real-world dataset SPAData [1] to fine-tune the pre-trained DRSformer for another 105k iterations. Below are the quantitative results in terms of PSNR (Details are provided in Table 2):
>
> | Methods            | Rain200L  | Rain200H  | Rain800   | DID-Data  | DDN-Data  | SPAData   |
> | ------------------ | --------- | --------- | --------- | --------- | --------- | --------- |
> | DRSformer w/o CoIC | 39.32     | 29.27     | 28.85     | 34.91     | 33.71     | 45.46     |
> | DRSformer w/ CoIC  | **39.70** | **30.31** | **29.73** | **35.02** | **33.94** | **46.03** |
>
> The results indicate that even with MoEs, DRSformer directly tuned on SPAData suffers heavy performance drops on Rain200L, Rain200H, and Rain800 datasets. However, **with the help of CoIC, DRSformer could achieve improvements of 1.04dB, 0.88dB, and 0.57dB on Rain200H, Rain800, and the real-world SPAData datasets.** This observation reveals that we could train a comprehensive model targeted at deraining both synthetic and real-world data using CoIC. Additionally, we also obtained a remarkable improvement in deraining ability on RealInt, displayed in Figure 4 and Figure 14. For your convenience, please review the results on our anonymous GitHub pages: https://anonymous.4open.science/r/CoIC-730F/
>
> > Novelty of the method may be limited. The paper needs to further explain the differences with recent contrastive learning-based methods
>
> Please note that our observation in Figure 1 reveals that learning joint rain-/detail-aware representations may help models learn better on mixed datasets. **This requires learning an encoder capable of discriminating different rain types, as well as perceiving background details. However, rain and background details are coupled in the rainy image, and it is challenging to find a suitable criterion to discriminate rain streaks from mixed datasets**. The contrastive learning method [2] mentioned by the reviewer attempts to learn the **degradation factor** from the degraded image, **where the background details are overlooked.** Moreover, [2] basically assumes that the degradations from two different images are different. This assumption may be inapplicable to mixed rainy datasets, where two images may share the same background (Rain200L and Rain200H) or show a similar rain pattern. Therefore, [2] cannot be applied to extract joint rain-/detail-aware representations from rainy input. In contrast, **we solve this problem by designing negative exemplars that contain different rain patterns and different background details.** To circumvent the issue that there is no well-defined criterion for defining "different" rains, we introduce a *rain layer bank* and regard the *most dissimilar rain* from the rain layer bank as different from the current rain. To clearly elaborate on the differences between the proposed contrastive learning strategy and existing strategies, we have provided an in-depth discussion in the section of *Introduction*, which better highlights our contributions.
>
> [1]. Tianyu Wang et al. "Spatial attentive single-image deraining with a high quality real rain dataset." in CVPR 2019.
>
> [2]. Longguang Wang, et al. "Unsupervised degradation representation learning for blind super-resolution." in CVPR 2021.

---

> > ### Author Response · Authors · 2023-11-20
> > **Response to Reviewer iZXv**
> >
> > > Increase in the number of parameters and changes in inference time
> >
> > We summarize the number of parameters, FLOPs, and inference time for BRN, RCDNet, DGUNet, IDT, and DRSformer in the table below (Also included in *Appendix A.3*):
> >
> > | Model name | input size | #p w/o CoIC (M) | #$\Delta$p  (M) | FLOPs w/o CoIC (G) | FLOPs w/CoIC (G) | Time w/o CoIC (ms) | Time w/ CoIC (ms) |
> > | ---------- | ---------- | :-------------: | --------------- | ------------------ | ---------------- | ------------------ | ----------------- |
> > | BRN        | 512x512    |      0.38       | 0.21            | 392.9              | 393.3            | 332.2$\pm$38.1     | 364.2$\pm$33.0    |
> > | RCDNet     | 512x512    |      2.98       | 2.11            | 389.0              | 389.5            | 351.6$\pm$0.2      | 384.8$\pm$2.2     |
> > | DGUNet     | 512x512    |      3.63       | 1.62            | 396.8              | 397.2            | 161.0$\pm$5.5      | 198.8$\pm$6.8     |
> > | IDT        | 128x128    |      16.42      | 2.53            | 7.3                | 7.6              | 59.8$\pm$2.7       | 83.2$\pm$10.5     |
> > | DRSformer  | 512x512    |      33.67      | 14.12           | 440.8              | 441.1            | 833.6$\pm$0.5      | 886.7$\pm$11.0    |
> >
> > Generally, **the increased parameters are related to the intrinsic architecture of models** since the proposed CoIC performs layer-wise modulation. **It is noteworthy that the performance improvement is not mainly from the increased parameters**. Note that IDT equipped with CoIC with fewer increased parameters can result in much better performance improvement when compared to DRSformer in Table 1, where more parameters are brought by CoIC for DRSformer. Moreover, the changes in FLOPs and inference time (about 30ms) are almost the same for all models, which means that **the additional changes are mainly from the encoder but not the modulation process.**

---

> > > ### Comment · Reviewer_iZXv · 2023-11-22
> > > **Thanks**
> > >
> > > Many Thanks. Fine-tuning a pre-trained DRSformer using SPAData may be not fair. This is my main concern. It may be more fair to train two models (w/o CoIC and w/ CoIC) from scratch on all the datasets.

---

> > > > ### Author Response · Authors · 2023-11-22
> > > > **Thanks for your feedback**
> > > >
> > > > Thanks for your reply. We believe that the comparison on **all six datasets** (five synthetic and one real-world) is fair. Please recall that in the *first period*, we train DRSformer w/o and w/ CoIC using mixed five synthetic datasets **from scratch** to obtain the pre-trained models. Then, we add SPAData to train these models for another *second period*. **Therefore, both DRSformer w/o and w/ CoIC are trained using all six datasets *from scratch* if we merge these two training periods**. Additionally, we compare the performance of these models **on all six datasets, not only on SPAData**. Thus, we assert that our comparison of DRSformer w/o and w/ CoIC on all six datasets is a fair assessment. **Moreover, we are willing to conduct the experiment suggested by the reviewer in the future**.

---

> > > > > ### Comment · Reviewer_iZXv · 2023-11-23
> > > > > **Thanks**
> > > > >
> > > > > Many Thanks. I mainly argue it would be more reasonable and appropriate to train two models (w/o CoIC and w/ CoIC) from scratch on all the datasets. I will carefully make the final decision based on the opinions of other reviewers.

---

> > > > > > ### Author Response · Authors · 2023-11-23
> > > > > > **Thanks for your suggestion**
> > > > > >
> > > > > > Thanks for your response. *We are currently training two models (w/o and w/ CoIC)* using all available datasets. Due to days of training, we might not be able to present the results before the discussion deadline. *If feasible, we will include this comparison in our manuscript in the future*. In our view, during the second stage, both models are trained using all datasets from a well-initialized state. **Even with the solid initialization, the model without CoIC still struggles to learn from the five synthetic datasets and the SPAData dataset**. Therefore, *it may be reasonable to deduce that the model without CoIC would face greater challenges when trained from scratch on synthetic and SPAData datasets*. **To sum up, we believe that the comparison on six datasets could already demonstrate the effectiveness of CoIC. However, as suggested by the reviewers, we will delve deeper into exploring its superiority**.

---

### Official Review · Reviewer_v7hb · 2023-11-05

**Soundness:** 3 good
**Presentation:** 2 fair
**Contribution:** 2 fair
**Rating:** 6
**Confidence:** 5

**Summary:**

This paper proposes a context-based instance-specific modulation (CoI-M) method for learning adaptive image de-raining models with mixed datasets. The goal is to exploit the commonalities and discrepancies among datasets for training. This mechanism can efficiently modulates both CNN and Transformer architectures. CoI-M is also verified to improve the performances of existing models when training on mixed datasets.

**Strengths:**

+ As to the paper structure, I think it is clear and easy to follow.

+ The authors claims that images with light rain are primarily characterized by background detail, while heavy rainy images are distinguished more by the rain itself. The statistical analysis in Figure 1 is suitable.

+ The experimental results indicate that the proposed method can further improve performance.

**Weaknesses:**

- It is not clear how to select the positive and negative exemplars. The author should discuss the differences with relevant contrastive learning-based deraining methods [1-2].
[1] Chen et al, Unpaired deep image deraining using dual contrastive learning, CVPR2022.
[2] Ye et al, Unsupervised deraining: Where contrastive learning meets self-similarity, CVPR2022

- I suggest the author conduct further validation on the mixed dataset Rain13K.

- The figures included in the manuscript are rendered with small font sizes and low resolution, which makes them challenging for reviewers to examine and comprehend. I recommend revising the figures to meet these requirements to improve the overall quality and accessibility of the visuals in the manuscript.

-----------------------After Rebuttal---------------------------

Thank you for your feedback. The rebuttal addressed my concerns well. I have decided to increase my score.

**Questions:**

See the above Weaknesses part.

---

> ### Author Response · Authors · 2023-11-20
> **Response to reviewer v7hb**
>
> **Comments**: Many thanks for your feedback and constructive suggestions. We are glad that the reviewer find out paper easy to follow and our statistical analysis is suitable. We will carefully address you issues below:
>
> > It is not clear how to select the positive and negative exemplars. Differences between the proposed method and relevant constrastive learning-based deraining methods should be detailed.
>
> We apologize for not clearly clarifying the differences between the proposed contrastive learning and existing contrastive learning methods. Our findings in Figure 1 suggest that learning joint rain-/detail-aware representations is meaningful for training models on mixed multiple datasets. This implies the need to train an encoder capable of discriminating different rains as well as background details. Hence, **the negative exemplars should contain exemplars with different rains and exemplars with different background details**. However, the recent method DCD-GAN [1] only considers the differences between rain and a clean background, which cannot perceive different rain types as well as background details. Additionally, the unsupervised method NLCL [2] also considers the differences between rain and a clean background. Although [2] introduces location contrastive learning to constrain the restored background to be the same as the ground truth, it cannot extract detail-aware information from the rainy input. Hence, [2] cannot be used to provide joint rain-/detail-aware guidance for adaptive image deraining. To clearly elaborate on the differences between our CoIC and recent contrastive learning methods, **we have added an in-depth discussion in the section of *Introduction***.
>
> We also trained the unsupervised method DCD-GAN on mixed synthetic datasets for more than 1M iterations. However, it failed to process intricate rains from multiple datasets. Below, we present its PSNR metrics (Details can be found in Table 1):
>
> | Methods          | Rain200L  | Rain200H  | Rain800   | DID-Data  | DDN-Data  |
> | ---------------- | --------- | --------- | --------- | --------- | --------- |
> | DCD-GAN          | 21.64     | 16.04     | 19.52     | 21.28     | 21.60     |
> | DRSformer + CoIC | **39.81** | **30.50** | **29.92** | **35.01** | **33.94** |
>
> In comparison to the proposed CoIC, DCD-GAN struggles on all synthetic benchmark datasets.
>
> > I suggest the author conduct further validation on the mixed dataset Rain13K.
>
> Thank you for this constructive suggestion. Since marginal improvement of CoIC on DRSformer has been observed in Table 1, it may be helpful to train it on another mixed dataset to validate its efficacy. However, as pointed out by reviewer nGEY, the Rain13K dataset could be regarded as one entity. Therefore, to further explore the potential of CoIC, **we have added a real-world dataset SPAData [3] to fine-tune DRSformer**. Surprisingly, **we observe that the tuned model using CoIC can achieve much better performance on both synthetic and real-world SPAData datasets**. We present the quantitative results in terms of PSNR below (Details are provided in Table 2):
>
> | Methods            | Rain200L  | Rain200H  | Rain800   | DID-Data  | DDN-Data  | SPAData   |
> | ------------------ | --------- | --------- | --------- | --------- | --------- | --------- |
> | DRSformer w/o CoIC | 39.32     | 29.27     | 28.85     | 34.91     | 33.71     | 45.46     |
> | DRSformer w/ CoIC  | **39.70** | **30.31** | **29.73** | **35.02** | **33.94** | **46.03** |
>
> Moreover, **the fine-tuned DRSformer with CoIC can efficiently remove complex real rain streaks in RealInt**. We have showcased high-quality real-rain removal results in Figure 4 and Figure 14. For your convenience, please review them on our anonymous GitHub page: https://anonymous.4open.science/r/CoIC-730F/.
>
> > The figures included in the manuscript are rendered with small font size and low resolution, which makes them challenging for reviewers to examine and comprehend. I recommend revising the figures to meet these requirements to improve the overall quality and accessibility of the visuals in the manuscript.
>
> We apologize for any inconvenience caused by the small font size in the figures, which may have posed readability issues. In the revised manuscript, we have adjusted the figures and increased the font size to enhance the reading experience.
>
> [1]. Xiang Chen, et al. "Unpaired deep image deraining using dual contrastive learning." In CVPR 2022.
>
> [2]. Yuntong Ye, et al. "Unsupervised deraining: Where contrastive learning meets self-similarity." in CVPR 2022.
>
> [3]. Tianyu Wang et al. "Spatial attentive single-image deraining with a high quality real rain dataset." in CVPR 2019.

---

### Official Review · Reviewer_zqPW · 2023-11-05

**Soundness:** 3 good
**Presentation:** 3 good
**Contribution:** 3 good
**Rating:** 6
**Confidence:** 5

**Summary:**

The paper focuses on tackling the single image deraining problem, wherein the authors introduce instance-specific de-raining models. These models are designed to delve into meaningful representations that capture the distinct characteristics of both the rainy elements and the background components in images affected by rain. Authors propose Context-based Instance-specific Modulation (CoI-M) mechanism which can modulate CNN- or Transformer-based to learn the representations specific to rain details and background. Authors also propose  rain-/detail-aware contrastive learning strategy to help extract joint rain-/detail-aware instance-specific representations. Integrating these modules authors claim that the proposed method can handle multiple different rain datasets.

**Strengths:**

- proposed Context-based Instance-specific Modulation (CoI-M) mechanism which can modulate CNN- or Transformer-based to learn the representations specific to rain details and background
     - employed feature extractor E,  and a Global Average Pooling (GAP) operation to capture rich spatial and channel information related to rain and image details.
- proposed contrastive learning strategy to help extract joint rain-/detail-aware instance-specific representations
     - to make the encoder discriminate the rain in (x,y) pair of rainy images they use leverage a rain layer bank noted a D_{R}
     - proposed contrastive learning based loss to learn the difference in representations of different rains and also image representations.
- Proposed Context-based Instance-aware Modulation (CoI-M) mechanism to modulate features at different layers in CNN, and attention layers in transformer based networks, to learn the embedding space spanned by all instance-specific representations.

**Weaknesses:**

- the comparisons are inadequate. There is rich literature in semi-supervised and continual learning that focuses on representation learning for different types of rain and image representations. Authors failed to compare with the existing methods.
    -  Memory Oriented Transfer Learning for Semi-Supervised Image Deraining, CVPR 2021.
        proposes a representation learning based approach where they learn basis vectors (which called memory) to represent the rain and adapt them using them for different datasets to minimize the differences the datasets. The proposed Col-C is also doing similar to this, so it will be easy for the reader to understand the benefits of CoI-C if authors can compare with this method.
    - Syn2Real Transfer Learning for Image Deraining using Gaussian Processes, CVPR 2020.
        proposes a Gaussian process based pesudo labeling approach where enable the encoder to learn representation rain and image using generated pesudo label. Note the here in Syn2Real they formulate the joint Gaussian distribution to generated a pesudo label for the unknown or unlabeled image, in way they selecting the k-nearest labeled images and maximizing correlation between unlabeled and labeled  similar type of rain images. Also minimizing correlation between k-nearest farthest labeled images and  unlabeled and labeled  similar type of rain images. Thus Syn2Real approach can be compared to proposed contrastive loss and CoI-C approach, to understand benefits of CoI-C.
     - Unpaired Deep Image Deraining Using Dual Contrastive Learning, CVPR 2022.
       proposed a contrastive based approach to learn the representation of rain and image and guide the networks to learn removal network and image generator networks in an unsupervised cyce-GAN approach. Thus, it would be great to see comparisons of CoI-C against this method.

**Questions:**

Please refer weaknesses

**Details Of Ethics Concerns:**

not aplicable

---

> ### Author Response · Authors · 2023-11-20
> **Response to reviewer zqPW**
>
> **Comments**: We appreciate the efforts and valuable suggestions provided by the reviewer. We will address the concerns outlined below:
>
> > the comparisons are inadequate. There is rich literature in semi-supervised and continual learning that focuses on representation learning for different types of rain and image representations. Authors failed to compare with the existing methods.
>
> We first added the semi-supervised method Syn2Real [1], MOSS [2], and the unsupervised method DCD-GAN [3] to our *Introduction* and literature review in *Related Work*. Since the official training code for MOSS [2] is unavailable, we compare the proposed CoIC with Syn2Real and DCD-GAN. Both Syn2Real and DCD-GAN are trained for more than 1M iterations (much longer than BRN, RCDNet, DGUNet, IDT, and DRSformer). However, we observed that both Syn2Real and DCD-GAN struggle when faced with intricate rainy images from multiple datasets, yielding unsatisfactory results on Rain200L, Rain200H, Rain800, DID-Data, and DDN-Data datasets. We tabulate the PSNR metrics below for direct comparison (Details are in Table 1 of the revised manuscript):
>
> | Methods          | Rain200L  | Rain200H  | Rain800   | DID-Data  | DDN-Data  |
> | ---------------- | --------- | --------- | --------- | --------- | --------- |
> | Syn2Real         | 30.83     | 17.21     | 24.85     | 26.71     | 29.15     |
> | DCD-GAN          | 21.64     | 16.04     | 19.52     | 21.28     | 21.60     |
> | DRSformer + CoIC | **39.81** | **30.50** | **29.92** | **35.01** | **33.94** |
>
> From the above results, we observe that learning a comprehensive deraining model on mixed datasets is challenging for semi-supervised and unsupervised methods. The proposed CoIC can efficiently assist models in learning much better on multiple datasets.
>
> Additionally, we have provided a demo on our anonymous GitHub page: https://anonymous.4open.science/r/CoIC-730F/
>
> [1]. Yasarla, Rajeev, et al. "Syn2real transfer learning for image deraining using gaussian processes." in CVPR 2020.
>
> [2]. Huaibo Huang, et al. "Memory oriented transfer learning for semi-supervised image deraining." in CVPR 2021.
>
> [3]. Xiang Chen, et al. "Unpaired deep image deraining using dual contrastive learning." In CVPR 2022.

---

> > ### Comment · Reviewer_zqPW · 2023-12-04
> >
> > Thankyou Authors for providing the answers for my concerns. After going through other reviewers feedback and authors feedback I updated my final rating.

---

### Official Review · Reviewer_nGEY · 2023-11-09

**Soundness:** 3 good
**Presentation:** 3 good
**Contribution:** 3 good
**Rating:** 5
**Confidence:** 5

**Summary:**

This paper proposes an instance-specific de-raining models by exploring representations that characterize both the rain and background components in rainy images. The authors first propose a rain-/detailaware contrastive learning strategy to explore joint rain-/detail-aware representations. Leveraging these representations as instructive guidance, the authors furhter introduce CoI-M to perform layer-wise modulation of CNNs and Transformers.

**Strengths:**

1, The paper is well-written.

2, The paper may be meaningful to some extent for the deraining community.

3, The results reveal that the proposed method consistently improves previous baselines.

4, The proposed instance-specific method may be useful to other low-level vision tasks, e.g., dehazing, desnowing, super-resolution, etc.

**Weaknesses:**

Although this paper has some strengths, I am still confused by following questions:

1, The proposed techniques seem to be a guidance for deraining, i.e., the authors attempt to constrain an instance-specific representation to guide the deraining.

Although this, the reviewer would like to know the increased parameters and FLOPs.

2, Whether the training iterations of the proposed method are the same with the applied method? i.e., BRN, RCDNet, DGUNet, IDT, DRSformer. If not, the comparisons are unfair.

3, The reviewers would like to see the training curve comparisons, e.g., BRN, RCDNet, DGUNet, IDT, DRSformer vs. BRN+ CoIC, RCDNet+ CoIC, DGUNet+ CoIC, IDT+ CoIC, DRSformer+ CoIC.

4, The paper title is the  INSTANCE-SPECIFIC IMAGE DE-RAINING, the reviewer thinks that this is not suitable since the authors also train the deep models on Rain13K. The dataset can be regarded as one entirety. It is hard to reflect the 'INSTANCE'. Whether it has a better title? The reviewer thinks that if the paper does a continual learning for deraining and demonstrates the CoIC can consistently improve the  performance on each dataset, the 'INSTANCE' may be suitable.

5, Deraining methods with high PSNR and SSIM usually tend to overfit to synthetic datasets. Hence, deraining performance would be worse on real datasets. Whether the authors can further solve this problem?

6, Can the rain-aware negative exemplars be images with real rain streaks? Whether the authors consider to explore the real images to participate in training to improve the generalization to real scenes?

7, I have observed that the authors mention the generalization many times. However, only training on synthetic dataset is hard to improve the generalization because of overfitting.

**Questions:**

See Weaknesses

**Details Of Ethics Concerns:**

None.

---

> ### Author Response · Authors · 2023-11-20
> **Response to reviewer nGEY (Part 1/2)**
>
> **Comment**: We thank the reviewer for the critical and constructive suggestions. We are delighted that you found our paper well-written and meaningful. However, there still exist critical issues that need to be addressed. Here, we will carefully address your concerns and questions:
>
> > About the increased parameters and FLOPs.
>
> As suggested by the reviewer, we have added a comprehensive analysis of CoIC in terms of the parameters (#P), FLOPs, and testing time. The results are presented in the table below. Generally, **the increased parameters are related to the intrinsic architecture of the models**. **It is noteworthy that the performance improvement is not mainly from the increased parameters**. Note that IDT equipped with CoIC, with fewer increased parameters, can result in much better performance improvement when compared to DRSformer in Table 1, where more parameters are brought by CoIC for DRSformer. However, the changes in FLOPs and inference time (about 30ms) are almost the same for all models, which means that the extra changes are mainly from the encoder but not the modulation process.
>
> We have also included this analysis in our paper. Please refer to *Appendix A.3* for details.
>
> | Model name | input size | #P w/o CoIC (M) | #$\Delta$P  (M) | FLOPs w/o CoIC (G) | FLOPs w/CoIC (G) | Time w/o CoIC (ms) | Time w/ CoIC (ms) |
> | ---------- | ---------- | :-------------: | --------------- | ------------------ | ---------------- | ------------------ | ----------------- |
> | BRN        | 512x512    |      0.38       | 0.21            | 392.9              | 393.3            | 332.2$\pm$38.1     | 364.2$\pm$33.0    |
> | RCDNet     | 512x512    |      2.98       | 2.11            | 389.0              | 389.5            | 351.6$\pm$0.2      | 384.8$\pm$2.2     |
> | DGUNet     | 512x512    |      3.63       | 1.62            | 396.8              | 397.2            | 161.0$\pm$5.5      | 198.8$\pm$6.8     |
> | IDT        | 128x128    |      16.42      | 2.53            | 7.3                | 7.6              | 59.8$\pm$2.7       | 83.2$\pm$10.5     |
> | DRSformer  | 512x512    |      33.67      | 14.12           | 440.8              | 441.1            | 833.6$\pm$0.5      | 886.7$\pm$11.0    |
>
> > Whether the training iterations of the proposed method are the same with the applied model?
>
> We apologize for missing the training details. In fact, to ensure a fair comparison, all the applied models were trained with consistent iterations and the pixel-fidelity loss. The training details are provided in *Appendix A.5*.
>
> > Comparison on the training curves.
>
> The training histories are visualized on our anonymous GitHub repository: https://anonymous.4open.science/r/CoIC-730F/. Additionally, we have included the training details and training history in *Appendix A.5*.

---

> > ### Author Response · Authors · 2023-11-20
> > **Response to reviewer nGEY (Part 2/2)**
> >
> > > The paper title is the **INSTANCE-SPECIFIC IMAGE DE-RAINING**, the reviewer thinks that this is not suitable since the authors also train the deep models on Rain13K. The dataset can be regarded as one entirety. It is hard to reflect the 'INSTANCE'. Whether it has a better title? The reviewer thinks that if the paper does a continual learning for deraining and demonstrates the CoIC can consistently improve the performance on each dataset, the 'INSTANCE' may be suitable.
> >
> > Thanks for this constructive comment. ***First***, we train models on the amalgamated datasets of Rain200L, Rain200H, Rain800, DID-Data, and DDN-Data. Our mixed dataset is *larger than the Rain13K* dataset with 28,900 examples. The intra-/inter-dataset similarity analysis in Figure 5(b) indicates that it *may be unsuitable to consider the whole dataset as an entity*. ***Second***, we acknowledge that it is hard to reflect the **INSTANCE** based on our existing experiments only on synthetic datasets. Therefore, we add a real-world dataset SPAData [1] to tune DRSformer w/o and w/ the proposed CoIC. Quantitative results are tabulated below (Details can be found in Table 2 of our revised manuscript):
> >
> > | Methods            | Rain200L  | Rain200H  | Rain800   | DID-Data  | DDN-Data  | SPAData   |
> > | ------------------ | --------- | --------- | --------- | --------- | --------- | --------- |
> > | DRSformer w/o CoIC | 39.32     | 29.27     | 28.85     | 34.91     | 33.71     | 45.46     |
> > | DRSformer w/ CoIC  | **39.70** | **30.31** | **29.73** | **35.02** | **33.94** | **46.03** |
> >
> > From the above results, we find that the proposed CoIC enables DRSformer to learn much better on both synthetic and real-world datasets (1.04 dB, 0.88 dB, and 0.57 dB improvement on Rain200H, Rain800, and SPAData, respectively). To clearly state our purpose and contribution, we have changed the title to: **Mixing and Modulating: Harnessing Joint Rain-/Detail-aware Representations to Eliminate Intricate Rains**.
> >
> > > Whether the author can further solve the worse deraining performance problem on real-world datasets?
> >
> > Current models usually struggle to perfectly derain real-world images when being trained on synthetic datasets. Thanks to the valuable comments above and from other reviewers, we find it simple yet effective to achieve **remarkable** real-world deraining ability by **adding a real-world dataset to tune the model**. Specifically, we tune DRSformer on the real-world dataset SPAData for about 105k iterations, and we have observed **significant real-world deraining performance improvement**. Please see our updated visualizations in Figure 4 and Figure 14. For convenience, we strongly recommend the reviewer to check our anonymous GitHub page on: https://anonymous.4open.science/r/CoIC-730F/.
> >
> > > Can the rain-aware negative exemplars be images with real rain streaks?
> >
> > There is no limitation on the rain-aware negatives. In practice, we tune the DRSformer on both synthetic and real-world SPA datasets for about 105k iterations, where synthetic and real-world rain streaks can fully interact through Equation 3. Surprisingly, we have attained remarkable real-world deraining ability on the SPAData dataset, which demonstrates the potential of the proposed method. Therefore, **the rain-aware negative exemplars can also be real rain streaks**.
> >
> > > I have observed that the authors mention the generalization many times. However, only training on synthetic dataset is hard to improve the generalization because of overfitting.
> >
> > To enhance the *real-world generalization ability*, we fine-tune the pre-trained DRSformer on the real-world SPA dataset. Experimental results demonstrate a significant improvement in real-world deraining ability. **In summary, the proposed CoIC enables us to train a powerful model capable of effectively deraining both synthetic and real-world rain scenarios simultaneously.**
> >
> > [1].  Tianyu Wang et al. "Spatial attentive single-image deraining with a high quality real rain dataset." in CVPR 2019.

---

### Author Response · Authors · 2023-11-19
**General Response to AC and Reviewers**

**Comment**: We appreciate the insightful feedback and valuable suggestions of all the reviewers. We have made appropriate changes to our manuscript based on the comments from reviewers. We have also conducted a variety of additional experiments to highlight the superiority of the proposed method and to address the concerns of reviewers. **Our new experiments tuning the model on a real-world dataset suggest that we may underestimate the potential of the proposed method**. Additionally, we have provided a real-world deraining demo and more high-quality real-world deraining results on the anonymous GitHub page: https://anonymous.4open.science/r/CoIC-730F/. And **we strongly recommend the reviewers to check our new impressive results**. We provide a brief summary of the additional experiments and manuscript changes.

### Experiments

- We provide the training loss curves for BRN, RCDNet, DGUNet, IDT, and DRSformer. Moreover, their complexities in terms of the number of parameters, FLOPs, and testing time are provided.
- We compare the proposed method with two extra representative methods: Syn2Real [1] and DCD-GAN [2] on mixed synthetic datasets.
- Instead of training models using the proposed CoIC on more synthetic datasets, we tune the pre-trained DRSformer on a real-world dataset SPAData [3]. The experimental results indicate that the proposed CoIC can offer significant performance improvement, *e.g.*, 1.04 dB, 0.88 dB, 0.57 dB on Rain200H, Rain800, and SPAData, respectively. These results demonstrate that **we could learn a comprehensive deraining model good at deraining both synthetic and real-world rains simultaneously.**

### Manuscript Changes

- We have changed the title of this manuscript to "**Mixing and Modulating: Harnessing Joint Rain-/Detail-aware Representations to Eliminate Intricate Rains**," which clearly highlights our purpose and contributions.

- We revised the *Introduction* (Paragraph 4) with an in-depth discussion on the limitations of previous contrastive learning strategies, as suggested by the reviewers, in order to underline our contributions.
- We have reviewed three additional methods in the *Related Work*: a heavy rain removal method [4], a semi-supervised method Syn2Real [1], and an unsupervised method DCD-GAN [2].
- We have added a literature review of "Image Restoration with Modulation" in the *Related Work* section.
- We have revised all figures with a larger font size to facilitate reading.
- We have added Syn2Real and DCD-GAN for comparison in *Section 4.2 Main Results*.
- We have substituted Figure 4 with high-quality real-world deraining results.
- We have added a subsection to explore the potential of CoIC when training models with both synthetic and real-world datasets. *See Section 4.2 Further Tuning on Real-world Dataset*.
- We moved the subsections "How does CoIC Modulate the Model" and "Balance Between Data-Fidelity and Contrastive Loss" to Appendix A.1 and Appendix A.4, respectively.
- We added model complexity analysis in Appendix A.3, model training details, and histories in Appendix A.5.
- We provide much better real-world deraining visualizations in Appendix A.9 (Also available in our demo https://anonymous.4open.science/r/CoIC-730F/ for convenience).

In our revised manuscript, all changes are highlighted in **magenta** for convenience.

We hope that the additional experimental results and changes could strengthen the state of our submission and address the concerns of the reviewers well. We look forward to further discussion.

[1]. Yasarla Rajeev, et al. "Syn2real transfer learning for image deraining using gaussian processes." in CVPR 2020.

[2]. Xiang Chen, et al. "Unpaired deep image deraining using dual contrastive learning." In CVPR 2022.

[3]. Tianyu Wang et al. "Spatial attentive single-image deraining with a high quality real rain dataset." in CVPR 2019.

[4]. Ruoteng Li et al. "Heavy rain image restoration: Integrating physics model and conditional adversarial learning." in CVPR 2019.

---

### Meta-Review · Area_Chair_s69n · 2023-12-11

**Metareview:**

This paper proposes a CoI-M method to learn adaptive image deraining models on mixed datasets. In addition, it introduces a joint rain-/detail aware contrastive learning strategy to help the estimation of CoI-M. The provided results show the effectiveness of the proposed methods.

The concerns of reviewers include inadequate comparisons, the way for selecting the positive and negative exemplars, necessarily of the proposed model, and limited novelty of the proposed method.

The authors solve the most concerns of reviewers. Based on the recommendations of reviewers, the paper can be accepted.

**Justification For Why Not Higher Score:**

N/A

**Justification For Why Not Lower Score:**

The major concerns from the most negative reviewer have been solved well. Especially, for the network parameters, FLOPs, training curves, novelty clarification.

Based on the recommendations of reviewers, the paper can be accepted. However, the generalization ability of the proposed should be better evaluated in the camera-ready paper.

---

### Decision · Program_Chairs · 2024-01-16

Accept (poster)